# TRAINING SPIKING NEURAL NETWORKS WITH REAL-TIME PROPAGATION THROUGH TIME

## ABSTRACT

Online learning algorithms for Spiking Neural Networks (SNNs) offer a memory-efficient alternative to Backpropagation Through Time (BPTT), but suffer from two critical issues: training instability and membrane potential distribution drift. To address these challenges, we introduce Real-Time Propagation Through Time (RPTT), a novel online learning framework. RPTT computes gradients using only the spatial component and integrates two synergistic regularization mechanisms: Membrane Potential Distribution Regularization (MPDR), which statistically constrains membrane potentials to counteract distributional drift, and Spatio-Temporal Gradient Regularization (STGR), which smooths weight updates to ensure stable convergence. We theoretically prove that RPTT converges to a stationary point. Extensive experiments on CIFAR-10/100, ImageNet-1k, and DVS-CIFAR10 demonstrate that RPTT achieves state-of-the-art performance while significantly reducing memory consumption. Experimental analysis reveals that RPTT achieves strong performance by effectively alleviating the membrane potential drift. Our work thus provides an effective framework for the online training of SNNs, significantly advancing their application in dynamic and realistic environments.

## 1 INTRODUCTION

Spiking Neural Networks (SNNs) are regarded as a new generation of efficient neural computing models. By transmitting and processing information through discrete spike events, they can handle temporal data and dynamic information more effectively, exhibiting dynamic characteristics that resemble those of biological neural systems (Ghosh-Dastidar & Adeli, 2009; Tavanaei et al., 2019). Since neurons incur computational and communication costs only when firing spikes, SNNs can achieve significantly lower energy consumption on neuromorphic hardware, offering a pronounced efficiency advantage over traditional Artificial Neural Networks (ANNs) (Yamazaki et al., 2022).

Nevertheless, training SNNs remains highly challenging. Firstly, the non-differentiability of the spiking function prevents the direct application of gradient-based optimization. In recent years, Backpropagation Through Time (BPTT) with Surrogate Gradients (SG) has become the dominant framework (Neftci et al., 2019), enabling end-to-end optimization and achieving competitive performance on complex tasks (Guo et al., 2023a). However, the computational cost of BPTT grows linearly with both network depth and sequence length. This results in excessive memory overhead, frequent gradient explosion or vanishing, and limited scalability to large-scale datasets such as ImageNet. Secondly, SNNs training is hampered by drift of the membrane potential distribution over time. Distribution drift refers to the tendency of the membrane potential distribution to deviate from the firing threshold. This phenomenon has been observed in networks trained with BPTT and originates from the intrinsic neuronal dynamics of SNNs (Guo et al., 2022b; 2023b; Liu et al., 2025). The resulting offset significantly degrades the network's performance.

To overcome the memory efficiency issue of BPTT, researchers have proposed various online learning algorithms (Guo et al., 2023a; Xiao et al., 2022; Meng et al., 2023). By computing local gradient approximations without storing full sequences, these methods can reduce memory costs. Existing online learning algorithms can be broadly categorized into two types: delayed update methods (Meng et al., 2023), which update parameters only after processing the entire sequence; and single update methods, which update parameters immediately at each time step. While delayed update

methods showed better stability, they fail to adapt promptly to the input distribution shifts in dynamic or non-stationary environments (Lobo et al., 2020), and lack biological plausibility (Bi & Poo, 1998). In contrast, single update methods enable real-time adaptation (Xiao et al., 2022; Bellec et al., 2020), making them appealing for dynamic and non-stationary environments. However, frequent updates also introduce noise, leading to unstable convergence (Kag & Saligrama, 2021; Wang et al., 2024; Cesa-Bianchi & Orabona, 2021).

While recent studies have investigated firing rate stability in online SNNs (Zhu et al., 2024), the specific issue of membrane potential distribution drift remains largely unexplored. Our work uniquely focuses on addressing this drift through online, layer-wise regularization. In fact, our preliminary analysis (detailed in Section 4.3, Figure 3 (b)) reveals that this problem is significantly more severe in the online setting compared to offline BPTT, as frequent parameter perturbations further amplify the errors introduced by approximate gradients. Such drift pushes the membrane potential distribution away from firing thresholds, weakening spiking activity and reducing representational capacity.

To address the membrane potential distribution drift and unstable convergence of SNNs online training, we propose a novel online learning algorithm, Real-Time Propagation Through Time (RPTT). Our algorithm updates parameters at each time step by computing only spatial gradients, and introduces two regularization terms: Membrane Potential Distribution Regularization (MPDR) and Spatio-Temporal Gradient Regularization (STGR). MPDR constrains per-layer membrane potential distribution with KL divergence and variance regularization toward a Gaussian target to effectively mitigating drift. STGR stabilizes weight updates through moving averages and suppresses noisy gradients via sparsity constraints. We theoretically proved that RPTT converges to stationary points of the empirical risk. Experimentally, we evaluated our approach on widely used static datasets and neuromorphic datasets. Our main contributions are summarized as follows:

- We propose RPTT, a memory-efficient online learning algorithm for SNNs that introduces two synergistic mechanisms: MPDR to counteract distributional drift and STGR to ensure stable convergence.
- We provide a theoretical analysis to prove the stability of our method and show that the parameter sequence generated by RPTT converges to a stationary point of the global empirical risk.
- We conduct extensive experiments on four challenging benchmarks, demonstrating that RPTT achieves state-of-the-art or highly competitive performance. Our analysis further reveals that online learning exacerbates membrane potential drift, an issue effectively mitigated by RPTT, which successfully stabilizes potential distributions and restores neuronal activity.

## 2 RELATED WORK

### 2.1 OFFLINE LEARNING ALGORITHMS IN SNNs

Training SNNs is inherently challenging due to their complex dynamics and the non-differentiability of spikes (Yamazaki et al., 2022). To address this, a prominent line of work utilizes well-trained ANNs and transfers their weights to SNNs, giving rise to the ANN-to-SNN method (Cao et al., 2015). Early studies primarily investigated how to switch between different types of neurons, proposing distinct conversion strategies (Diehl et al., 2015; Rueckauer et al., 2017). Subsequent efforts have largely focused on reducing conversion loss to match the accuracy of ANNs (Bu et al., 2023; 2022; Wang et al., 2022). However, these improvements often require a large number of inference time steps and thus fail to exploit the intrinsic temporal dynamics of SNNs (Ding et al., 2021). Direct training approaches adopt SG techniques to enable BPTT in deep SNNs, allowing training with only a few time steps (Guo et al., 2023a). This line of work has fostered the independent development of SNNs beyond ANNs conversion, with advances in neuron models and network architectures achieving performance comparable to ANNs in certain tasks. However, the scalability of BPTT remains limited by heavy memory consumption and gradient vanishing (Guo et al., 2024), and it deviates from the biological principle of synapses being updated in a timely manner (Lillicrap & Santoro, 2019). Several studies have addressed performance degradation in SNNs arising from temporal drift in the membrane potential distribution. For instance, Guo et al. (2022b) observed

this drift during spike propagation and proposed penalizing statistical deviations to prevent degradation and saturation. Building on this, Liu et al. (2025) further analyzed the drift across time steps, mitigating it with an adaptive surrogate function and additional supervisory signals.

## 2.2 ONLINE LEARNING ALGORITHMS IN SNNS

Online learning algorithms for SNNs compute gradients in each time step, thereby avoiding the long-term dependencies inherent in BPTT and achieving better efficiency in both memory and computation (Lobo et al., 2020). Early research on online learning algorithms was primarily conducted in the domain of Recurrent Neural Networks (RNNs). Williams & Zipser (1989) introduced Real-Time Recurrent Learning (RTRL), which performs online gradient computation by maintaining a Jacobian matrix of weights. Subsequently, Bellec et al. (2020) proposed e-prop, which combines eligibility traces with learning signals to achieve more efficient online computation. Building on this idea, Xiao et al. (2022) proposed Online Training Through Time (OTTT), which adopts eligibility traces while discarding gradients through the reset mechanism, thereby enabling online training in SNNs. Jiang et al. (2024) further developed Neuronal Dynamics-based Online Training (NDOT), which provides a more fine-grained modeling of intra-layer temporal dependencies within the OTTT framework to improved performance. These methods all employ a single update strategy. In contrast, Meng et al. (2023) proposed Spatial Learning Through Time (SLTT), which adopts a delayed update strategy and discards temporal gradients, computing only spatial gradients to achieve memory-efficient training. Moreover, to address the unavailability of future gradient information, Zhu et al. (2024) proposed the OSR and OTS modules, which ensure consistent normalization parameters and stabilize neuronal firing rates across time steps, respectively. However, the issue of unstable convergence due to frequent parameter updates in online training has received little attention. Meanwhile, frequent parameter updates exacerbate membrane potential distribution shift over time steps, impairing SNNs performance.

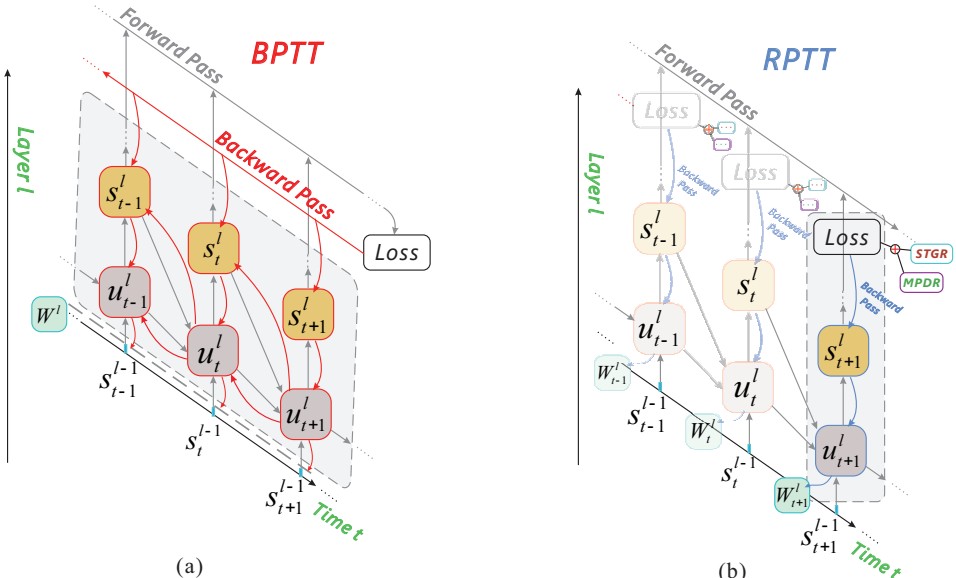

Figure 1: (a) The forward and backward procedures of BPTT. (b) The forward and backward procedures of RPTT.

## 3 METHOD

### 3.1 SNNS MODEL AND LEARNING ALGORITHM

Spiking neurons mimic the behavior of biological neurons. They integrate incoming spikes to update their membrane potential, fire a spike upon exceeding a threshold, and then reset the potential. This dynamic is formally described by the Leaky Integrate-and-Fire (LIF) model, whose commonly used discrete form is given as follows:

$$\begin{cases} u_i[t+1] = (1 - \frac{1}{\tau_m})(u_i[t] - s_i[t]V_{th}) + \sum_j w_{ij}s_j[t] + b_i, \\ s_i[t+1] = H(u_i[t+1] - V_{th}), \end{cases} \quad (1)$$

where $u_i[t]$ is the membrane potential at step $t$ by neuron $i$, $s_i[t]$ denotes the spike train, $V_{th}$ is the firing threshold, $\tau_m$ represent the membrane time constant. The weight from neuron $j$ to neuron $i$ is $w_{ij}$, and $b_i$ is a bias. $H(x)$ is the Heaviside step function, which is usually approximated by the SG method. BPTT is commonly used for direct training of SNNs, and its expansion equation is as follows:

$$\frac{\partial \mathcal{L}}{\partial \mathbf{W}^l} = \sum_{t=1}^{T} \frac{\partial \mathcal{L}}{\partial \mathbf{s}^{l+1}[t]} \frac{\partial \mathbf{s}^{l+1}[t]}{\partial \mathbf{u}^{l+1}[t]} \left( \frac{\partial \mathbf{u}^{l+1}[t]}{\partial \mathbf{W}^l} + \sum_{\tau < t} \prod_{i=t-1}^{\tau} \left( \frac{\partial \mathbf{u}^{l+1}[i+1]}{\partial \mathbf{u}^{l+1}[i]} + \frac{\partial \mathbf{u}^{l+1}[i+1]}{\partial \mathbf{s}^{l+1}[i]} \frac{\partial \mathbf{s}^{l+1}[i]}{\partial \mathbf{u}^{l+1}[i]} \right) \frac{\partial \mathbf{u}^{l+1}[\tau]}{\partial \mathbf{W}^l} \right), \quad (2)$$

where $\mathbf{W}^l$ is the weight from layer $l$ to $l + 1$, $\mathcal{L}$ is the loss function, and $T$ is the total time step size. Due to the non differentiability of $\frac{\partial \mathbf{s}^{l+1}[t]}{\partial \mathbf{u}^{l+1}[t]}$, derivatives of rectangular functions $\frac{\partial s}{\partial u} = \frac{1}{a_c}\text{sign}\left(|u - V_{th}| < \frac{a_c}{2}\right)$ often used as a substitute ($a_c$ is a constant), among others. In classification tasks using SNNs, assuming the network has $N$ layers, the network outputs are given by $\mathbf{o}[t] = \mathbf{W}^o\mathbf{s}^N[t]$, where $\mathbf{W}^o$ is the classifier's weights, $\mathbf{s}^N[t]$ is the output spike trains of the $N$-th layer. The classification is based on the average of these outputs across all time steps, computed as $\frac{1}{T}\sum_{t=1}^{T}\mathbf{o}[t]$. BPTT loss function $\mathcal{L}$ is often defined as $\mathcal{L} = \ell(\frac{1}{T}\sum_{T}^{t=1}\mathbf{o}_t, \mathbf{y})$, where $\mathbf{y}$ is the label, and $\ell$ can be the cross-entropy function. BPTT introduces multiplicative dependencies across time, leading to high memory demands and making it prone to gradient vanishing or explosion when applied over long sequences. In contrast, online learning algorithms alleviate this problem by performing backpropagation at each individual time step (Xiao et al., 2022; Bellec et al., 2020). In this setting, a local loss function $\ell_t$ is defined at each time step, and the gradients of certain variables in the BPTT formulation are simplified or approximated. Two update strategies are commonly used in online learning algorithms: The single update strategy computes gradients and updates parameters at each time step, while the delayed update strategy postpones parameter updates until the end of the time step.

## 3.2 REAL-TIME PROPAGATION THROUGH TIME FOR SNNs

In this section, we present the single update online learning algorithm, Real-Time Propagation Through Time (RPTT). RPTT calculates gradients using the spatial component of BPTT and incorporates two additional regularization terms: Membrane Potential Distribution Regularization (MPDR) and Spatio-Temporal Gradient Regularization (STGR) (Figure 1 (b)). Figure 1 (a) illustrates the computational graph of BPTT, which has to maintain the computational graph of previous time to backpropagate through time. To enable online computation, Our approach computes gradients at each time step using only the spatial component by discarding temporal backpropagation. The choice of using spatial gradients is motivated by the observation that the temporal component contributes only marginally to the gradient in BPTT, which was also reported in the delayed update online learning algorithm SLTT (Meng et al., 2023). Therefore, RPTT discarded the temporal components $\sum_{\tau < t} \prod_{i=t-1}^{\tau} \left( \frac{\partial \mathbf{u}^{l+1}[i+1]}{\partial \mathbf{u}^{l+1}[i]} + \frac{\partial \mathbf{u}^{l+1}[i+1]}{\partial \mathbf{s}^{l+1}[i]} \frac{\partial \mathbf{s}^{l+1}[i]}{\partial \mathbf{u}^{l+1}[i]} \right) \frac{\partial \mathbf{u}^{l+1}[\tau]}{\partial \mathbf{W}^l}$, and we have:

$$\frac{\partial \mathcal{L}}{\partial \mathbf{W}^l} = \sum_{t=1}^{T} \left( \frac{\partial \mathcal{L}}{\partial \mathbf{s}^{l+1}[t]} \frac{\partial \mathbf{s}^{l+1}[t]}{\partial \mathbf{u}^{l+1}[t]} \frac{\partial \mathbf{u}^{l+1}[t]}{\partial \mathbf{W}^l} + a_1 \frac{\partial \mathcal{L}_M^l[t]}{\partial \mathbf{W}^l} + a_2 \frac{\partial \mathcal{L}_S^l[t]}{\partial \mathbf{W}^l} \right), \quad (3)$$

where $\mathcal{L}_M^l[t]$ and $\mathcal{L}_S^l[t]$ are the MPDR and STGR terms (see Equation 4 and 5), respectively. $a_1$ and $a_2$ are constants. To achieve the instantaneous loss calculation of $\ell_t = \frac{1}{T}\ell(\mathbf{o}[t], \mathbf{y})$, we adopt the loss function $\mathcal{L} = \frac{1}{T}\sum_{T}^{t=1}\ell(\mathbf{o}[t], \mathbf{y})$, which is an upper bound of the loss introduced in the BPTT loss. Note that in our online learning framework, the network parameters are updated at every time step. Therefore, we use the index $t$ to denote both the simulation time step and the optimization iteration, i.e., $\mathbf{W}_t$ represents the weights at time step $t$.

**MPDR**: Frequent parameter updates can intensify the inherent membrane potential distribution drift in SNNs. This drift can impair performance. We analyze this in detail in the experimental Section 4.3. To address this issue, we introduce the Membrane Potential Distribution Regularization (MPDR). The central idea is to add a statistical constraint on the distribution of membrane potentials at each layer. This constraint enforces the potentials to remain within a predefined, biologically inspired distribution (Rudolph et al., 2004), thereby mitigating the performance degradation caused by membrane potential distribution drift. The MPDR is defined as follows:

$$\mathcal{L}_M^l[t] = KL(p_t^l(V_t) \| q^l(V_t)) + \mu Var(V_t^l), \tag{4}$$

here, $p_t^l(V_t)$ denotes the empirical distribution of membrane potentials in the $l$-th layer at time step $t$, while $q^l(V_t)$ represents the target distribution of each layer (assumed to be Gaussian). $Var(V_t^l)$ penalizes excessive variance, thereby stabilizing the membrane potential distribution. The hyperparameter $\mu$ controls the strength of this term.

**STGR**: Due to frequent parameter updates, single update methods inject stochastic gradient noise at each step while tracking a drifting optimum. This process leads to a parameter error composed of both bias and variance, which cannot be minimized simultaneously. To address this issue, we introduce Spatio-Temporal Gradient Regularization (STGR). STGR explicitly enforces smoothness between successive parameter states to alleviate rapid fluctuations, while penalizing large gradients to encourage sparsity and prevent instability. This formulation is given as follows:

$$\mathcal{L}_S^l[t] = \frac{\lambda}{2}\|\mathbf{W}_t^l - \hat{\mathbf{W}}_t^l\|^2 + \frac{\gamma}{2}\|\nabla \ell_{t-1}(\mathbf{W}_t^l)\|^2, \tag{5}$$

$$\hat{\mathbf{W}}_{t+1}^l = \rho\hat{\mathbf{W}}_t^l + (1-\rho)\mathbf{W}_{t+1}^l, \rho \in (0,1), \tag{6}$$

let $\mathbf{W}_t^l$ denote the weight matrix of the $l$-th layer at time step $t$, and $\hat{\mathbf{W}}_t^l$ denote the moving average of $\mathbf{W}_t^l$. The coefficient $\lambda$ controls the first term. $\nabla \ell_{t-1}(\mathbf{W}_t^l)$ represents the gradient from the previous time step. This selectively suppresses gradient noise generated during silent periods of neurons, improving the signal-to-noise ratio and overall training efficiency. The global scaling factor $\gamma$ represents the contribution of this term. Our RPTT process as detailed in Algorithm 1.

---

**Algorithm 1** One iteration of SNNs training with the RPTT

---

**Input:** Training data $(\mathbf{s}^0, \mathbf{y})$; Time steps $T$; Learning rate $\eta$; Hyper-parameters $a_1$, $a_2$, $\rho$
**Initialize:** $\mathbf{W}^l$ randomly in the domain $\mathcal{W}$
**for** $t = 1, 2, ..., T$ **do**
    Update $\mathbf{u}^l[t]$ and $\mathbf{s}^l[t]$
    Update $\ell_t(\mathbf{W}^l) = \frac{1}{T}\ell(\mathbf{o}[t], \mathbf{y})$
    Calculate $\ell'(\mathbf{W}^l) = \ell_t(\mathbf{W}^l) + a_1\mathcal{L}_M^l[t] + a_2\mathcal{L}_S^l[t]$
    **for** $l = N, N-1, ..., 1$ **do**
        Calculate $\nabla_{\mathbf{W}^l}\ell'(\mathbf{W}_t^l) = \frac{\partial \ell_t(\mathbf{W}^l)}{\partial \mathbf{W}^l} + a_1\frac{\partial \mathcal{L}_M^l[t]}{\partial \mathbf{W}^l} + a_2\frac{\partial \mathcal{L}_S^l[t]}{\partial \mathbf{W}^l}$
        Update $\mathbf{W}_{t+1}^l = \mathbf{W}_t^l - \eta\nabla_{\mathbf{W}^l}\ell'(\mathbf{W}^l)\big|_{\mathbf{W}^l = \mathbf{W}_t^l}$
        Update $\hat{\mathbf{W}}_{t+1}^l = \rho\hat{\mathbf{W}}_t^l + (1-\rho)\mathbf{W}_{t+1}^l$
    **end for**
**end for**
**Output:** $\{\mathbf{W}_T^l\}_{l=1}^N$

---

### 3.3 CONVERGENCE THEORY OF RPTT

In this section, we will prove that the weight sequence $\{\mathbf{W}_t^l\}$ generated by RPTT, converges to the stationary point $\mathbf{W}^*$ under the update rule Equation 6. To simplify the description, remove the layer superscripts, and we assume that all layers share the same target membrane potential distribution $q(V)$, but the analysis can be extended in a layer-wise manner without changing the convergence result.

**Lemma 1:** Fixed target distribution $q(V) = \mathcal{N}(\mu^*, \sigma^{*2})$, defined as $S_t$ as follows:

$$S_t = KL(\mathcal{N}(\mu_t, \sigma_t^2) \| \mathcal{N}(\mu^*, \sigma^{*2})) + Var(V_t), \tag{7}$$

there exists a constant $K > 0$ with $S_t \leq K$ for all $t$. Assumption and proof details can be found in the Appendix A.1.1.

**Lemma 2:** Consider the following form of inequality:

$$\sum_{t=1}^{T} \eta_t \|\nabla \ell_t(\mathbf{W}_t)\|^2 \leq B_0 + h \sum_{t=1}^{T} \eta_t \|\nabla \ell_{t-1}(\mathbf{W}_{t-1})\|^2 + B_T, \tag{8}$$

here, $\ell_t$ is instantaneous loss and $\eta_t$ is learning rate at time step $t$, $B_0$ is a constant, $B_T$ is bounded with $T$, and $0 < h < 1$, when $T \to \infty$, then:

$$\sum_{t=1}^{\infty} \eta_t \|\nabla \ell_t(\mathbf{W}_t)\|^2 \leq \frac{A + h\eta_1 \|\nabla \ell_0(\mathbf{W}_0)\|^2 + \sup_T B_T}{1 - h} < \infty, \tag{9}$$

assumption and proof details can be found in the Appendix A.1.1.

**Theorem 1:** We assume that the loss function $\ell_t(\cdot)$ is $\beta$- smooth, with a lower bound of $L_{\inf}$, and that the task sequence changes satisfy $|\ell_{t+1}(\mathbf{W}) - \ell_t(\mathbf{W})| \leq \Delta_t$ and $\sum_{t=1}^{\infty} \Delta_t < \infty$. The remaining technical assumptions can be found in Appendix A.1.1. Then the square weighted average of the gradient converges to zero:

$$\lim_{T \to \infty} \frac{\sum_{t=1}^{T} \eta_t \|\nabla \ell_t(\mathbf{W}_t)\|^2}{\sum_{t=1}^{T} \eta_t} = 0, \tag{10}$$

this means that there exists a subsequence $t_k$ such that $\|\nabla \ell_{t_k}(\mathbf{W}_{t_k})\| \to 0$, and $\mathbf{W}_{t_k}$ converges to the stationary point $\mathbf{W}^*$ of the global empirical risk $\mathcal{L}$, i.e., $\nabla \mathcal{L}(\mathbf{W}^*) = 0$. We leverage the $\beta$-smoothness property of the loss function and apply Young's inequality to handle gradient cross terms, transforming the convergence problem into proving $\sum_{t=1}^{\infty} \eta_t \|g_t\|^2 < \infty$. By analyzing the boundedness of MPDR and STGR and absorbing them into the main convergence analysis, we ultimately establish that the weighted average of squared gradients converges to zero, ensuring algorithmic convergence to a stationary point. Please refer to the Appendix A.1.1 for the detailed proof.

## 4 EXPERIMENTS

We evaluate our approach on widely used datasets (CIFAR-10, CIFAR-100, ImageNet-1k, DVS-CIFAR10), and compare against state-of-the-art SNNs training algorithms. Furthermore, we conduct layer-wise analysis of membrane potentials as well as ablation studies on the proposed MPDR and STGR components.

### 4.1 COMPARISON WITH BPTT

**Experimental setup.** We trained both BPTT and RPTT under the same settings and analyzed their memory efficiency. To facilitate a fine-grained analysis of the memory consumption of MPDR and STGR, we additionally introduced an online learning algorithm with the single update strategy, Online Space Backpropagation (OSBP). This algorithm adheres to the configuration outlined in Section 3.2, utilizing only the spatial component for gradient computation and updating parameters at each time step.

**Memory Efficiency.** In terms of memory efficiency, our method's memory usage is independent of the number of time steps, whereas BPTT's memory consumption increases with the number of time steps (Figure 2 (a)). Conventional BPTT requires maintaining the full computational graph over the

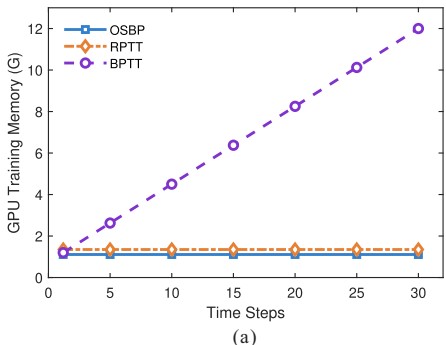 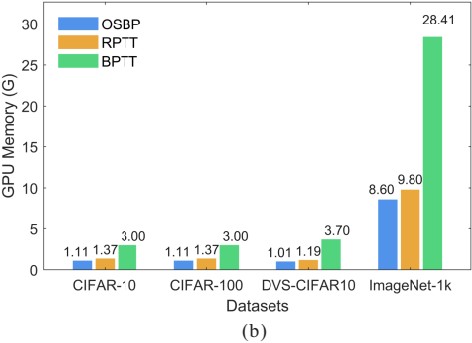

Figure 2: (a) Memory usage with the timestep $T$ on CIFAR-10. (b) Memory usage of different datasets at $T = 6$.

entire temporal sequence, leading to memory consumption that grows linearly with the number of time steps, i.e., $O(TN)$. In contrast, RPTT performs backpropagation independently at each time step without retaining cross-temporal graphs, resulting in a memory complexity of $O(N)$. We also report the memory consumption of different algorithms at $T = 6$ across four datasets (Figure 2 (b)). These results indicate that RPTT saves over 50% of memory consumption compared to BPTT. Compared with OSBP, RPTT incurs only a slight increase in memory usage, indicating that the additional cost introduced by the regularization terms is negligible.

### 4.2 COMPARISON WITH OTHER METHODS

We benchmark our method against other competitive algorithms of SNNs training. To further demonstrate that RPTT maintains competitive accuracy and is more stable on different datasets, we conducted experiments using the OSBP under the same settings as RPTT. The results are shown in Table 1. For the experiments on ImageNet-1K, the RPTT models were fine-tuned from SLTT-based pretrained checkpoints.

Compared to single update online learning algorithms, our method achieves competitive performance across all datasets. Specifically, RPTT achieves 95.17% and 75.62% accuracy on CIFAR-10 and CIFAR-100, respectively. On ImageNet-1k and DVS-CIFAR10, our method achieves 66.19% and 78.00%, demonstrating strong performance comparable to or exceeding other leading methods. Relative to OTTT (Xiao et al., 2022), RPTT reduces the per-epoch memory usage from 1.71 GB to 1.37 GB, as it discards the temporal component of gradient computation and avoids the additional memory required by OTTT to maintain eligibility traces.

Our method consistently outperforms delayed update online algorithms such as SLTT (Meng et al., 2023) in accuracy across all datasets. SLTT's performance benefits from postponing parameter updates; for instance, it outperforms OSBP, which also uses spatial gradients but updates parameters at each time step. In contrast, RPTT achieves superior results by integrating powerful regularization techniques while retaining the key advantages of a single-step update. Specifically, MPDR effectively mitigates membrane potential distribution drift to maintain neuronal activity, while STGR stabilizes the optimization trajectory against the noise inherent in frequent updates. Moreover, the two components of our RPTT framework, MPDR and STGR, can be readily incorporated into other online learning algorithms of SNNs without altering their overall structure.

### 4.3 ABLATION STUDY AND MEMBRANE POTENTIAL DISTRIBUTION ANALYSIS

**Ablation study.** To evaluate RPTT, we conduct four ablation studies on CIFAR-10: (i) OSBP, (ii) OSBP with STGR, (iii) OSBP with MPDR, and (iv) OSBP with both STGR and MPDR (i.e., RPTT). Under identical baseline parameters, the accuracy rates for these four experiments are 93.72%, 94.10%, 94.43%, and 95.17%, respectively (Figure 3 (a)). The results show that both STGR and MPDR individually yield notable accuracy improvements over the OSBP, while their combination achieves the best overall performance shown in Figure 3 (a). However, the contribution of STGR to accuracy improvement is smaller than that of MPDR, since STGR is primarily designed to stabilize the training process rather than directly enhance accuracy (see Appendix A.1.2 for a discussion of

Table 1: Comparison results with different methods on CIFAR-10 and CIFAR-100, ImageNet-1k and DVS-CFIAR10. The "Update" column means that the algorithm updates parameters at each time step.

| Dataset | Method | Network | T | Update | Accuracy |
|---|---|---|---|---|---|
| CIFAR-10 | OTTT$_O$ (Xiao et al., 2022) | VGG-11 (sWS) | 6 | ✓ | 93.49% |
| | OSBP | ResNet-18 | 6 | ✓ | 93.72% |
| | TET (Deng et al., 2022) | ResNet-19 | 6 | × | 94.50% |
| | SLTT (Meng et al., 2023) | ResNet-18 | 6 | × | 94.44% |
| | SLTT+STGR+MPDR | ResNet-18 | 6 | × | 94.68% |
| | NDOT$_O$ (Jiang et al., 2024) | VGG-11 (sWS) | 6 | ✓ | 94.89% |
| | SLOT (Xue et al., 2024) | ResNet-18 | 6 | ✓ | **95.47**% |
| | RPTT (Ours) | ResNet-18 | 6 | ✓ | 95.17% |
| CIFAR-100 | OTTT$_O$ (Xiao et al., 2022) | VGG-11 (sWS) | 6 | ✓ | 71.05% |
| | OSBP | ResNet-18 | 6 | ✓ | 74.59% |
| | SLTT (Meng et al., 2023) | ResNet-18 | 6 | × | 74.38% |
| | SLOT (Xue et al., 2024) | ResNet-18 | 6 | ✓ | 74.71% |
| | TET (Deng et al., 2022) | ResNet-19 | 6 | × | 74.72% |
| | NDOT$_O$ (Jiang et al., 2024) | VGG-11 (sWS) | 6 | ✓ | **76.61%** |
| | RPTT (Ours) | ResNet-18 | 6 | ✓ | 75.62% |
| ImageNet-1k | OTTT$_O$ (Xiao et al., 2022) | NF-ResNet-34 | 6 | ✓ | 64.16% |
| | SLOT (Xue et al., 2024) | Sew ResNet-34 | 6 | ✓ | 64.90% |
| | TET (Deng et al., 2022) | Spiking-ResNet-19 | 6 | × | 64.79% |
| | SLTT (Meng et al., 2023) | NF-ResNet-34 | 6 | × | 66.19% |
| | RPTT (Ours) | NF-ResNet-34 | 6 | ✓ | **66.19%** |
| DVS-CIFAR10 | OSBP | VGG-11 | 10 | ✓ | 76.90% |
| | OTTT$_O$ (Xiao et al., 2022) | VGG-11 (sWS) | 10 | ✓ | 76.63% |
| | SLTT (Meng et al., 2023) | VGG-11 | 10 | × | 77.17% |
| | TET (Deng et al., 2022) | VGGSNN | 10 | × | 77.33% |
| | NDOT$_O$ (Jiang et al., 2024) | VGG-11 (sWS) | 10 | ✓ | 77.50% |
| | RPTT (Ours) | VGG-11 | 10 | ✓ | **78.00%** |

STGR's stabilizing effect). This is supported by Figure 3 (d), which shows that the variance across layers is consistently lower when STGR is applied compared to OSBP. This reduction in variance indicates that STGR effectively dampens the optimization oscillations caused by frequent updates, thereby fostering a more stable training process. In contrast, MPDR is specifically designed to maintain a well-structured membrane potential distribution, which in turn reduces information loss and ultimately improves network performance. The combination of the two achieves the best overall results, indicating that the two regularization terms are complementary.

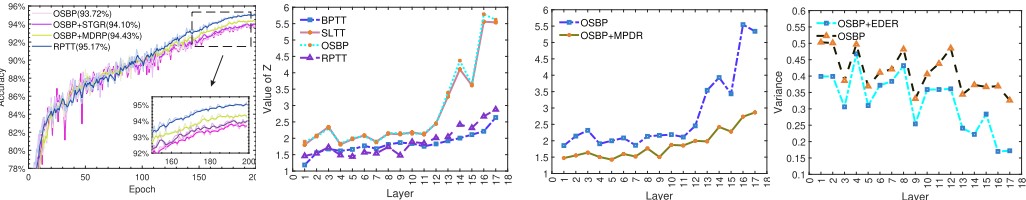

Figure 3: (a) Accuracy curves under different experiments. (b) Under the same experimental setup, we show the layer-wise $Z_l$ variation at the first time step for ResNet-18 trained with BPTT, SLTT, and RPTT on CIFAR-10. (c) The $Z_l$ of different layers for OSBP and OSBP with MPDR in ResNet-18. (d) The variance of different layers for OSBP and OSBP with STGR in ResNet-18.

**Membrane potential distribution analysis.** The phenomenon of membrane potential distribution drift over time has been observed in offline algorithms (Liu et al., 2025). In this paper, we quantify the shift of membrane potential distributions to demonstrate that the drift problem becomes more severe under online training. Since the membrane potential distribution can be approximated by a

Gaussian, we adopt the $Z$-score to measure the deviation of the distribution from the threshold. The $Z$-score for the $l$-th layer is given by $Z_l = (V_{th} - \mu_l)/\sigma_l$. We report the $Z$-score trajectories for BPTT, SLTT, and RPTT in Figure 3 (b). To ensure a fair comparison within the online learning framework, we adapted SLTT to perform real-time updates. Compared with the offline algorithm BPTT, the online learning algorithm SLTT exhibits more severe drift, with its $Z$-score exceeding BPTT at all layers. This effect can be attributed to challenges inherent to online training. Specifically, frequent parameter updates and approximation errors in gradient computation hinder neurons from maintaining an optimal membrane potential distribution. In contrast, the $Z$-score under RPTT is comparable to that of BPTT, because MPDR constrains the membrane potential distribution and thereby reduces drift in the context of online training. Furthermore, to validate the effectiveness of our proposed component, we compare the $Z$-score of OSBP and OSBP with MPDR in Figure 3 (c). The results show that the inclusion of MPDR significantly lowers the $Z$-score, effectively alleviating the drift. This demonstrates that the MPDR component plays a crucial role in maintaining a stable membrane potential distribution.

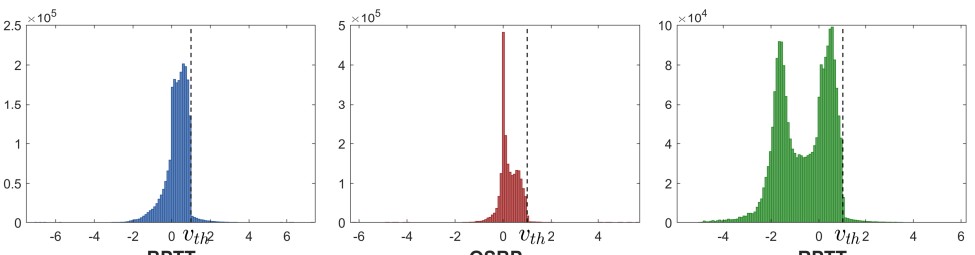

Figure 4: The membrane potential distribution of BPTT, OSBP, and RPTT, with $v_{th}$ as the threshold and the vertical axis as the probability density. The input data is from CIFAR-10, with the same fixed samples, and the neurons are taken from the first layer of ResNet-18.

Interestingly, we also observed a bimodal distribution in the first layer of networks trained with RPTT (Figure 4), whereas deeper layers evolved toward a Gaussian distribution (please refer to the Appendix A.2.5 for detailed analysis). This indicates that the regularization encourages a functional differentiation among neurons, where part of the population is suppressed to low membrane potentials while another part remains close to the firing threshold. Such a separation effectively prevents all neurons from clustering around the threshold and enhances coding diversity. This bimodal distribution is also present in the membrane potential of layer 2/3 pyramidal neurons in the somatosensory barrel cortex of anesthetized rats, where it reflects the network's dynamic transitions between low- and high-activity states (Petersen et al., 2003). Similar observations have also been reported in prior work of SNNs. For example, Guo et al. (2022a) employed soft-reset IF neurons with a Membrane Potential Rectifier (MPR). This method transformed the membrane potential distribution from Gaussian to bimodal, thereby reducing information loss from quantization errors. Unlike MPR, which modifies the neuronal mechanisms, our method tackles the problem from the perspective of alleviating membrane potential distribution drift. Interestingly, this also leads to the emergence of a bimodal distribution, suggesting that our MPDR functions similar to MPR which mitigates information loss during spike propagation.

## 5 CONCLUSION

In this paper, we addressed the critical challenges of convergence instability and membrane potential drift that hinder online learning in SNNs. We proposed Real-Time Propagation Through Time (RPTT), a memory-efficient algorithm that backpropagates using spatial gradients and incorporates two novel regularization techniques: Spatio-Temporal Gradient Regularization (STGR) to ensure stable learning dynamics and Membrane Potential Distribution Regularization (MPDR) to directly counteract distributional drift. Our theoretical analysis guarantees the convergence of RPTT to a stationary point. Our extensive experiments confirm the effectiveness of RPTT, which achieves state-of-the-art performance on multiple benchmarks. Crucially, our analysis shows that RPTT mitigates the severe membrane potential drift observed in other online methods, an advantage that is particularly pronounced in deeper network layers. The core mechanisms of RPTT, MPDR and STGR, offer

a generalizable framework for enhancing other online SNNs algorithms. This work thus advances the feasibility of deploying powerful neuromorphic systems in real-time, resource-constrained environments.

**Reproducibility Statement.** We are committed to ensuring the reproducibility of our research. All datasets used in this work (CIFAR-10/100, ImageNet-1k, and DVS-CIFAR10) are publicly available and standard benchmarks in the field. The core of our proposed method, RPTT, including its components MPDR and STGR, is detailed in Section 3.2, with a step-by-step implementation outlined in Algorithm 1. The complete theoretical analysis and convergence proofs are provided in Appendix A.1.1. All experimental settings, network architectures, and hyperparameters are thoroughly described in Appendix A.2. While the source code is not included with this initial submission, we pledge to release a fully documented, open-source implementation upon publication to facilitate further research and verification by the community.

**Ethics Statement.** This work adheres to the ICLR Code of Ethics. The research was conducted with a commitment to scientific integrity, ensuring transparency in our methods and the honest reporting of results to promote reproducibility. All datasets used in this study are publicly available, and we have respected their licensing and privacy terms. We have considered the potential societal impacts of our work and believe it contributes positively to the responsible development of machine learning.

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

# A APPENDIX

**LLM Usage Statement.** We utilized a large language model (LLM) as a writing assistance tool during the preparation of this manuscript. The primary uses of the LLM were for grammar checking across the entire paper and for polishing the language and improving the clarity of the abstract. We have carefully reviewed and edited all text, including LLM-generated suggestions, to ensure scientific accuracy and take full responsibility for the final content of this paper.

This section provides detailed theoretical analyses, experimental settings, and additional visualizations. A.1.1 presents the complete proof of the convergence theory in Section 3.3, together with the proofs of the two supporting lemmas, while Appendix A.1.2 offers a theoretical explanation of the stability of STGR. Appendix A.2 reports the experimental details, including the datasets (A.2.1), normalization strategies (A.2.2), network architectures (A.2.3), training hyperparameters (A.2.4), as well as additional visualization and STGR stability experiments (A.2.5).

## A.1 DETAILED DERIVATION AND PROOFS

### A.1.1 CONVERGENCE THEORY

We now proceed to the proofs of Theorem 1, Lemma 1, and Lemma 2. To simplify the description, we do not consider the number of samples, remove the layer superscripts of all symbols, and we assume that all layers share the same target membrane potential distribution $q = \mathcal{N}(\mu^*, \sigma^{*2})$, but the analysis can be extended in a layer-wise manner without changing the convergence result, simplified symbol $p_t$ denotes the distribution of membrane potentials at time step $t$.

**Marking and updating equations:**

For the convenience of presentation, we have changed $\mathbf{W_t}$ to $W_t$.

Task gradient: $g_t := \nabla \ell_t(W_t)$.

STGR gradient $u_t := \nabla \mathcal{L}_S[t]$.

MPDR gradient $v_t := \nabla \mathcal{L}_M[t]$.

Total gradient $G_t := g_t + a u_t + b v_t$.

Updating equations $W_{t+1} = W_t - \eta_t G_t$, $\hat{W}_{t+1} = \rho \hat{W}_t + (1-\rho) W_{t+1}$, where $\rho \in (0,1)$.

**Assumptions:**

**1.** Assume that $\ell_t(.)$ is $\beta$-smooth with respect to $W$ and has a lower bound $\ell_t(W) \geq L_{\inf}$, and for all $W$, $|\ell_{t+1}(W) - \ell_t(W)| \leq \Delta_t$, with $\sum_{t=1}^{\infty} \Delta_t < \infty$.

**2.** Assume the gradient is bounded, i.e., $\exists G > 0, \forall t, \|\nabla \ell_t(W)\| \leq G$. Additionally, assume the membrane potential distribution follows $\mathcal{N}(\mu_t, \sigma_t^2)$, and $\exists \sigma_{\min} > 0, \forall t, \sigma_t \geq \sigma_{\min}$ when approximating $p_t$. For all layer, assume membrane potential $V_t \in [V_{min}, V_{max}]$.

**3.** Assume the learning rate $\eta_t > 0$ is non-increasing and satisfies $\sum_{t=1}^{\infty} \eta_t = \infty$, $\sum_{t=1}^{\infty} \eta_t^2 < \infty$, and $\sum_{t=1}^{\infty} \eta_t^3 < \infty$.

**Justification of Assumptions:**

**1.** It ensures the smoothness and stability of the loss sequence.

**2.** It can be satisfied in practice by gradient clipping (to guarantee bounded gradients), the empirical observation that membrane potentials approximately follow Gaussian distributions, explicit bounding of $V_t$ in the model design, and variance truncation to enforce $\sigma_t \geq \sigma_{min}$.

**3.** It is standard in stochastic approximation and can be satisfied, for instance, by setting $\eta_t = \Theta(1/t^{e_1})$ with $e_1 \in (1/2, 1]$.

**Theorem 1:** Under the above Assumptions, the square weighted average of the gradient converges to zero:

$$\lim_{T \to \infty} \frac{\sum_{t=1}^{T} \eta_t \|\nabla \ell_t(W_t)\|^2}{\sum_{t=1}^{T} \eta_t} = 0,$$

this means that there exists a subsequence $\{t_k\}$ such that $\|\nabla \ell_{t_k}(W_{t_k})\| \to 0$, and $\{W_{t_k}\}$ converges to the stationary point $W^*$ of the global empirical risk $\mathcal{L}$, i.e., $\nabla \mathcal{L}(W^*) = 0$.

**Proof:**

Assuming $\beta$-smooth given by $\ell_t$, for any $x, y$, there is

$$\ell_t(y) \leq \ell_t(x) + <\nabla \ell_t(x), y - x> + \frac{\beta}{2}\|y - x\|^2, \tag{1}$$

substituting $x = W_t, y = W_{t+1} = W_t - \eta_t G_t$, we have:

$$\ell_t(W_{t+1}) \leq \ell_t(W_t) - \eta_t < g_t, G_t > + \frac{\beta}{2}\eta_t^2\|G_t\|^2, \tag{2}$$

using Young's inequality for cross terms:

$$\langle g_t, u_t \rangle \geq -\frac{1}{2}\|g_t\|^2 - \frac{1}{2}\|u_t\|^2, \tag{3}$$

similarly, for $\langle g_t, v_t \rangle$, the same processing can be applied, and by substituting them, the lower bound can be obtained:

$$\langle g_t, G_t \rangle \geq (1 - \frac{a+b}{2})\|g_t\|^2 - \frac{a}{2}\|u_t\|^2 - \frac{b}{2}\|v_t\|^2, \tag{4}$$

among them, we can make $a + b < 2$.

Based on the definitions of STGR and MPDR, as well as Assumption 2, it can be concluded that there exist constants $C_u$ and $C_v$ such that:

$$\|u_t\|^2 \leq C_u(\|W_t - \hat{W}_t\|^2 + \|g_{t-1}\|^2), \tag{5}$$

$$\|v_t\|^2 \leq C_v(KL(p_t\|q) + Var(V_t)), \tag{6}$$

where the constant $C_u$ is determined by $\lambda, \gamma, V_{min}, V_{max}$.

Substitute (4) into (2):

$$\ell_t(W_{t+1}) \leq \ell_t(W_t) - \eta_t\left((1 - \frac{a+b}{2})\|g_t\|^2 - \frac{a}{2}\|u_t\|^2 - \frac{b}{2}\|v_t\|^2\right) + \frac{\beta}{2}\eta_t^2\|G_t\|^2. \tag{7}$$

Substitute (5) and (6) into (7), collect all non principal terms into $R_t$, and based on the assumption of gradient boundedness, there exists a constant upper bound $C$ such that:

$$\ell_t(W_{t+1}) \leq \ell_t(W_t) - (1 - \frac{a+b}{2})\eta_t\|g_t\|^2 + \eta_t R_t + C\eta_t^2, \tag{8}$$

among them, $R_t$ is:

$$R_t = \alpha_1\|W_t - \hat{W}_t\|^2 + \alpha_2\|g_{t-1}\|^2 + \alpha_3(KL(\rho_t\|q) + Var(V_t)), \tag{9}$$

where $\alpha_i > 0$ and $C > 0$ are given by $a$, $b$, $\beta$, $C_u$, and $C_v$.

Sum up (8) from $t = 1$ to $T$, by Assumption 1, we have:

$$\left(1 - \frac{a+b}{2}\right)\sum_{t=1}^{T}\eta_t\|g_t\|^2 \leq \ell_1(W_1) - \ell_T(W_{T+1}) + \sum_{t=1}^{T-1}[\ell_{t+1}(W_{t+1}) - \ell_t(W_{t+1})] + \sum_{t=1}^{T}\eta_t R_t + C\sum_{t=1}^{T}\eta_t^2$$

$$\leq \ell_1(W_1) - \ell_T(W_{T+1}) + \sum_{t=1}^{T-1}\Delta_t + \sum_{t=1}^{T}\eta_t R_t + C\sum_{t=1}^{T}\eta_t^2$$

$$\leq \ell_1(W_1) - L_{\inf} + \sum_{t=1}^{T-1}\Delta_t + \sum_{t=1}^{T}\eta_t R_t + C\sum_{t=1}^{T}\eta_t^2. \tag{10}$$

By $\sum_{t=1}^{T-1}\Delta_t < \infty$ and $\sum_{t=1}^{T}\eta_t^2 < \infty$, we just need to explain $\sum_{t=1}^{T}\eta_t R_t < \infty$, this formula is as follows:

$$\sum_{t=1}^{T}\eta_t R_t = \alpha_1\sum_{t=1}^{T}\eta_t\|W_t - \hat{W}_t\|^2 + \alpha_2\sum_{t=1}^{T}\eta_t\|g_{t-1}\|^2 + \alpha_3\sum_{t=1}^{T}\eta_t(KL(\rho_t\|q) + Var(V_t)), \tag{11}$$

for $\sum_{t=1}^{T}\eta_t\|W_t - \hat{W}_t\|^2$ in (11), we define $d_t := W_t - \hat{W}_t$, and we have:

$$d_{t+1} = \rho d_t + \rho(W_{t+1} - W_t) = \rho d_t - \rho\eta_t G_t, \tag{12}$$

applying the Young's inequality, for any $\epsilon > 0$, we have:

$$\|d_{t+1}\|^2 \leq \rho^2(1 + \epsilon)\|d_t\|^2 + \rho^2(1 + \frac{1}{\epsilon})\eta_t^2\|G_t\|^2, \tag{13}$$

given the arbitrariness of $\epsilon$, it is known that $\epsilon$ exists such that $\rho^2(1 + \epsilon) \in (0, 1)$.

Multiplying $\eta_{t+1}$ on both sides of (12) simultaneously, according to $\eta_{t+1} \leq \eta_t$, there are:

$$\eta_{t+1}\|d_{t+1}\|^2 \leq \rho^2(1 + \epsilon)\eta_t\|d_t\|^2 + \rho^2(1 + \frac{1}{\epsilon})\eta_t^3\|G_t\|^2, \tag{14}$$

iterating on (13) yields:

$$\eta_{t+1}\|d_{t+1}\|^2 \leq (\rho^2(1 + \epsilon))^t\eta_t\|d_t\|^2 + \rho^2(1 + \frac{1}{\epsilon})\sum_{k=1}^{t}(\rho^2(1 + \epsilon))^{t-k}\eta_k^3\|G_k\|^2, \tag{15}$$

sum $t$ on both sides and exchange the sum order:

$$\sum_{t=1}^{T} \eta_{t+1}\|d_{t+1}\|^2 \leq \frac{1}{1-\rho^2(1+\epsilon)}\eta_1\|d_1\|^2 + \frac{\rho^2(1+\frac{1}{\epsilon})}{1-\rho^2(1+\epsilon)}\sum_{k=1}^{T}\eta_k^3\|G_k\|^2, \qquad (16)$$

when $T \to \infty$, if $\sum \eta_t^3 < \infty$ and $\|G_t\|$ under bounded conditions, the right-hand side of the formula is finite, that is:

$$\sum_{t=1}^{\infty} \eta_t\|d_t\|^2 < \infty. \qquad (17)$$

For $\sum_{t=1}^{T} \eta_t\|g_{t-1}\|^2$ in (11), the non-increasing $\eta_t$ yields:

$$\sum_{t=1}^{T} \eta_t\|g_{t-1}\|^2 = \eta_1\|g_0\|^2 + \sum_{t=2}^{T}\eta_t\|g_{t-1}\|^2 \leq \eta_1\|g_0\|^2 + \sum_{t=2}^{T}\eta_{t-1}\|g_{t-1}\|^2 \leq \eta_1\|g_0\|^2 + \sum_{s=1}^{T-1}\eta_s\|g_s\|^2, \qquad (18)$$

let $T \to \infty$, we get:

$$\sum_{t=1}^{\infty} \eta_t\|g_{t-1}\|^2 \leq \eta_1\|g_0\|^2 + \sum_{t=1}^{\infty}\eta_t\|g_t\|^2. \qquad (19)$$

For $\sum_{t=1}^{T} \eta_t(KL(\rho_t\|q) + Var(V_t))$ in (11), according to Lemma 1 (Lemma 1 is provenn in 27), there is $S_t \leq K$ ($S_t$ is defined in 27). When $T \to \infty$ is present, the regularization term attenuation coefficient $c_t$ is introduced (which can be implemented in the code), and substituted into the equation:

$$\sum_{t=1}^{\infty} \eta_t c_t S_t \leq K\sum_{t=1}^{\infty}\eta_t c_t, \qquad (20)$$

from the assumption: $\eta_t = d_\eta/t^{e_1}$, $c_t = c_0/t^{e_2}$, where $d_\eta$ is the initial learning rate and $c_0$ is the initial regularization term strength, then:

$$\sum_{t=1}^{\infty} \eta_t c_t = d_\eta c_0 \sum_{t=1}^{\infty}\frac{1}{t^{e_1+e_2}}, \qquad (21)$$

this is a $p$-series that converges if and only if $e_1 + e_2 > 1$. Given the assumed step size decay exponent $e_1 \in (1/2, 1]$, choosing $e_2 > 1 - e_1$ ensures the convergence of $\sum_{t=1}^{\infty}\eta_t c_t$, meaning $\sum_{t=1}^{\infty}\eta_t c_t S_t$ is bounded. This holds under the constraint of the decay coefficient $c_t$ (for simplicity of presentation, the subsequent terms omit $c_t$, but this does not affect the results.).

$$\sum_{t=1}^{\infty} \eta_t S_t < \infty. \qquad (22)$$

Rewrite equation (10), since $\ell_t(W)$ is bounded, $\ell_1(W_1) - \ell_{T+1}(W_{T+1})$ has a constant upper bound $C_0$. Therefore, there exist constants $C_1, C_2, C_3, C_4$ such that:

$$\sum_{t=1}^{T} \eta_t \|g_t\|^2 \le \frac{2}{2-(a+b)} (\ell_1(W_1) - \ell_{T+1}(W_{T+1}) + \sum_{t=1}^{T} \eta_t R_t + C \sum_{t=1}^{T} \eta_t^2)$$

$$\le C_0 + \frac{2C}{2-(a+b)} \sum_{t=1}^{T} \eta_t^2 + \frac{2\alpha_1}{2-(a+b)} \sum_{t=1}^{T} \eta_t \|W_t - \hat{W}_t\|^2 \tag{23}$$

$$+ \frac{2\alpha_2}{2-(a+b)} \sum_{t=1}^{T} \eta_t \|g_{t-1}\|^2 + \frac{2\alpha_3}{2-(a+b)} \sum_{t=1}^{T} \eta_t S_t,$$

substitute (19) into the right-hand side and rearrange the terms (see the Lemma 2 in 31 for details):

$$(1 - \frac{2\alpha_2}{2-(a+b)}) \sum_{t=1}^{T} \eta_t \|g_t\|^2 \le C_0 + \frac{2C}{2-(a+b)} \sum_{t=1}^{\infty} \eta_t^2 + \frac{2\alpha_1}{2-(a+b)} \sum_{t=1}^{\infty} \eta_t \|W_t - \hat{W}_t\|^2$$

$$+ \frac{2\alpha_3}{2-(a+b)} \sum_{t=1}^{\infty} \eta_t S_t + \frac{2\alpha_2}{2-(a+b)} \eta_1 \|g_0\|^2,$$

$$\tag{24}$$

from (17), (22), and the assumptions, the right-hand side of (24) is finite. As $T \to \infty$, and with the condition $(1 - \frac{2\alpha_2}{2-(a+b)}) < 1$, we obtain:

$$\sum_{t=1}^{\infty} \eta_t \|g_t\|^2 < \infty, \tag{25}$$

that is:

$$\frac{\sum_{t=1}^{T} \eta_t \|g_t\|^2}{\sum_{t=1}^{T} \eta_t} \xrightarrow{T \to \infty} 0 \tag{26}$$

**lemma1:** Fixed target distribution $q = \mathcal{N}(\mu^*, \sigma^{*2})$, define:

$$S_t = KL\left(\mathcal{N}(\mu_t, \sigma_t^2) \| \mathcal{N}(\mu^*, \sigma^{*2})\right) + Var(V_t). \tag{27}$$

under Assumption 2, there exists a constant $K > 0$ (depending only on $V_{\min}, V_{\max}, \sigma_{\min}, \mu^*$, etc.) such that for all $t$, $S_t \le K$.

**Proof.** For any real-valued random variable $X$ on the interval $[a, b]$, the standard inequality holds:

$$\mathrm{Var}(X) \le \frac{(b-a)^2}{4}. \tag{28}$$

From Assumption 2, the variance of a single layer satisfies $\mathrm{Var}(V_t) \le \frac{(V_{max}-V_{min})^2}{4}$. If there are $N$ layers, the sum of the variances has an upper bound of $N \cdot \frac{(V_{max}-V_{min})^2}{4}$. If $p_t = \mathcal{N}(\mu_t, \sigma_t^2)$, $q = \mathcal{N}(\mu^*, \sigma^{*2})$, then:

$$KL(p_t \| q) = \ln\frac{\sigma^*}{\sigma_t} + \frac{\sigma_t^2 + (\mu_t - \mu^*)^2}{2\sigma^{*2}} - \frac{1}{2}. \tag{29}$$

By Assumption 2, the sample mean $\mu_t \in [V_{min}, V_{max}]$, hence $(\mu_t - \mu^*)^2 \le M_\mu$ (where $M_\mu$ is a constant). Furthermore, we have $\sigma_t^2 \le (V_{max} - V_{min})^2/4$. Since $\sigma_t \ge \sigma_{min} > 0$, it follows that $|\ln(\sigma^*/\sigma_t)| \le \max\{|\ln(\sigma^*/\sigma_{min})|, |\ln(\sigma^*/\sigma_{max})|\}$, where $\sigma_{max} = \sqrt{(V_{max} - V_{min})^2/4}$.

Therefore, the single-layer KL divergence has a constant upper bound, denoted as $C_{KL}$. Combining the variance upper bound and the KL upper bound, we set:

$$K = N \cdot \left( \frac{(V_{max} - V_{min})^2}{4} + C_{KL} \right), \tag{30}$$

$S_t \leq K$ holds for all $t$.

**Lemma 2:** From equation (19), the following absorption lemma can be derived. For the inequality

$$\sum_{t=1}^{T} \eta_t \|g_t\|^2 \leq B_0 + h \sum_{t=1}^{T} \eta_t \|g_{t-1}\|^2 + B_T, \tag{31}$$

where $B_0$ is a constant, $B_T$ is bounded with respect to $T$, and $0 \leq h < 1$. Substituting (19) into the right-hand side and rearranging terms yields

$$(1 - h) \sum_{t=1}^{T} \eta_t \|g_t\|^2 \leq B_0 + h\eta_1 \|g_0\|^2 + B_T. \tag{32}$$

Thus, we have:

$$\sum_{t=1}^{\infty} \eta_t \|g_t\|^2 \leq \frac{B_0 + h\eta_1 \|g_0\|^2 + \sup_T B_T}{1 - h} < \infty. \tag{33}$$

Note: The historical gradient penalty coefficient in STGR (along with the constant derived from smoothness) ensures that the resulting $h$ satisfies $h < 1$, allowing this lemma to be applied.

### A.1.2 PROOF OF THE STABILITY EFFECT OF STGR

SNNs exhibit complex dynamics due to nonlinear membrane potential evolution, discontinuous spiking events, and frequent parameter updates in online learning. These factors make it intractable to derive a strict convergence proof under the full nonconvex dynamics. Therefore, to facilitate the analysis, we adopt a quadratic approximation of the loss landscape in the neighborhood of the current parameter point. This approximation allows us to explicitly characterize the influence of the regularization term on parameter updates and stability.

We aim to show that the proposed regularization term

$$\mathcal{L}_S[t] = \frac{\lambda}{2} \|W_t - \hat{W}_t\|^2 + \frac{\gamma}{2} \|\nabla \ell_{t-1}(W_t)\|^2 \tag{34}$$

improves the stability of the online training process.

**Proof.** Without regularization, the standard online gradient update is

$$W_{t+1} = W_t - \eta \nabla \ell_t(W_t), \tag{35}$$

where $\eta$ is the learning rate and $\ell_t$ is the instantaneous loss function. Including the regularization term, the effective objective becomes

$$\tilde{L}_t(W) = \ell_t(W) + \mathcal{L}_S[t], \tag{36}$$

and its gradient at $W_t$ is

$$\nabla \tilde{L}_t(W_t) = \nabla \ell_t(W_t) + \lambda \left( W_t - \hat{W}_t \right) + \gamma H_{t-1} \nabla \ell_{t-1}(W_t), \tag{37}$$

where $H_{t-1} = \nabla^2 \ell_{t-1}(W)\big|_{W=W_t}$ denotes the Hessian of $\ell_{t-1}$ at $W_t$.

Using the quadratic approximation of $\ell_t$ around $W_t$,

$$\ell_t(W) \approx \ell_t(W_t) + \nabla \ell_t(W_t)^\top (W - W_t) + \frac{1}{2}(W - W_t)^\top H_t (W - W_t), \tag{38}$$

with $H_t = \nabla^2 \ell_t(W)\big|_{W=W_t}$, the update with regularization can be written as

$$W_{t+1} = W_t - \eta\Big(\nabla\ell_t(W_t) + \lambda(W_t - \hat{W}_t) + \gamma H_{t-1}\nabla\ell_{t-1}(W_t)\Big). \tag{39}$$

The stability of the update depends on whether the update operator contracts towards the local optimum. The term $\lambda(W_t - \hat{W}_t)$ acts as a quadratic penalty that suppresses oscillations in parameter space, while the term $\gamma H_{t-1}\nabla\ell_{t-1}(W_t)$ damps fluctuations caused by noisy online gradients.

Formally, in the local quadratic approximation, the effective curvature matrix becomes

$$H_t + \lambda I + \gamma H_{t-1}^2,$$

which increases the strong convexity constant of the model. A larger strong convexity constant implies a smaller condition number, thereby improving stability and reducing sensitivity to stochastic gradient noise. This analysis is based on local quadratic approximations and should be regarded as a heuristic justification rather than a rigorous proof. In addition, Appendix A.2.5 provides stability experiments of STGR under multiple random seeds, which empirically demonstrate that incorporating STGR leads to consistently smaller standard deviations.

## A.2 IMPLEMENTATION DETAILS

### A.2.1 DATASETS

We evaluate our model on both static and neuromorphic datasets. The static datasets include CIFAR-10, CIFAR-100, and ImageNet-1k, while DVS-CIFAR10 serves as the neuromorphic benchmark.

**CIFAR-10.** The CIFAR-10 dataset is a widely used benchmark in computer vision (Krizhevsky et al. (2009)), consisting of 60,000 color images with a spatial resolution of 32×32 pixels. The dataset is divided into 50,000 training images and 10,000 test images, evenly distributed across 10 object categories such as airplanes, automobiles, and animals. Its small image size and diverse classes make it a standard testbed for evaluating image classification models.

**CIFAR-100.** The CIFAR-100 dataset shares the same image format and size as CIFAR-10 but is more challenging due to its larger label space (Krizhevsky et al. (2009)). It contains 60,000 images in total, with 100 fine-grained classes, each comprising 600 images. The dataset is split into 50,000 training images and 10,000 test images, and its fine-level categorization provides a more difficult benchmark for generalization and scalability studies.

**ImageNet-1k.** ImageNet-1k is a large-scale benchmark dataset for image recognition (Deng et al. (2009)), containing approximately 1.28 million training images and 50,000 validation images across 1,000 object categories. Each image has high visual diversity, varying in resolution, viewpoint, background, and intra-class variation. Due to its scale and complexity, ImageNet-1k has been a primary benchmark for advancing deep learning in computer vision.

**DVS-CIFAR10.** The DVS-CIFAR10 dataset is an event-based version of CIFAR-10 (Li et al. (2017)), generated using a Dynamic Vision Sensor (DVS). Instead of conventional frames, it represents visual input as asynchronous spike events triggered by changes in luminance. The dataset contains event streams corresponding to the 10 CIFAR-10 categories, enabling evaluation of neuromorphic and spiking neural network algorithms under temporally sparse and bio-inspired data representations.

### A.2.2 SCALED WEIGHT STANDARDIZATION AND NF-RESNETS

An important characteristic of our RPTT framework is that gradients are computed instantaneously at each time step, which ensures memory costs remain independent of the total number of time steps. However, this property makes conventional batch normalization (BN) (Deng et al. (2022); Li et al. (2021); Meng et al. (2022); Zheng et al. (2021)) along the temporal dimension inapplicable, since BN requires collecting statistics across all time steps during the forward pass, incurring large memory overhead and breaking the online gradient calculation scheme. To address this issue, as done in other online learning algorithms such as OTTT (Xiao et al. (2022)), SLTT (Meng et al. (2023)), and NDOT (Jiang et al. (2024)), we adopt normalization strategies that are compatible with our framework: for some tasks, we calculate BN statistics independently at each time step, while for

others we follow the normalization-free paradigm (NF-ResNets) and replace BN with scaled weight standardization (sWS) (Brock et al. (2021a;b); Qiao et al. (2019)), which standardizes weights directly and avoids reliance on temporal statistics. The sWS component Brock et al. (2021a), modified from the original weight standardization (Qiao et al. (2019)), normalizes the weights according to the following formula:

$$\bar{W}_{i,j} = \gamma_s \frac{W_{i,j} - \mu_{W_{i,*}}}{\sigma_{W_{i,*}}}, \tag{40}$$

here, the mean $\mu_{W_{i,*}}$ and standard deviation $\sigma_{W_{i,*}}$ are computed across the fan-in dimension indexed by $i$. The hyperparameter $\gamma_s$ is adjusted to ensure stable signal propagation during the forward pass, with a value of $\gamma_s \approx 2.74$. For deep ResNets (He et al. (2016a)), the signal-preserving property of BN is not well maintained by sWS due to the skip connections. To address this, we adopt normalization-free ResNets (NF-ResNets) (Brock et al. (2021a;b)), which replace BN with sWS and incorporate additional design choices to stabilize the forward signal. Specifically, NF-ResNets use residual blocks of the form $x_{l+1} = x_l + \alpha_r f_l(x_l/\beta_l)$, where $\alpha_r$ and $\beta_l$ are hyperparameters used to stabilize signals, and the weights in $f_l(\cdot)$ are imposed with sWS. The weights are standardized by sWS, ensures unit variance, and $\alpha_r$ controls variance growth (set to 0.2 in our experiments).

### A.2.3 NETWORK ARCHITECTURE

For experiments on CIFAR-10 and CIFAR-100, we adopt ResNet-18 (He et al. (2016a)) with pre-activation residual blocks (He et al. (2016b)). The four residual stages contain 64, 128, 256, and 512 channels, respectively. All ReLU activations are replaced with Leaky Integrate-and-Fire (LIF) neurons, and max pooling layers are substituted with average pooling to ensure neuromorphic implementability. To support instantaneous gradient updates, batch normalization (BN) is applied independently at each time step rather than across the entire temporal horizon, resulting in $T$ BN operations per iteration, where $T$ is the number of time steps. For experiments on DVS-CIFAR10, we adopt VGG-11 (Simonyan & Zisserman (2014)) with the same modifications as ResNet-18, including average pooling and time-step-wise BN. Two fully connected layers (Deng et al. (2022); Meng et al. (2022); Xiao et al. (2022)) are removed to reduce computational cost, and dropout (Srivastava et al. (2014)) is applied after each LIF layer for regularization, with dropout rates set to 0.3. For ImageNet experiments, we adopt NF-ResNet-34, which are normalization-free ResNets.

### A.2.4 TRAINING SETTINGS AND HYPERPARAMETERS

**Training settings.** All implementations are developed using the PyTorch (Paszke et al. (2019)) and SpikingJelly (Fang et al. (2023)) frameworks, CIFAR-10, CIFAR-100, ImageNet-1k train on NVIDIA GeForce RTX 4090 GPU, and DVS-CIFAR10 train on NVIDIA GeForce RTX 3090 GPU. For ImageNet-1k, due to constraints on computational resources, we utilized a pre-trained SLTT model (Meng et al. (2023)), with the RPTT algorithm applied solely for fine-tuning. This is because SLTT is also an online learning algorithm that relies solely on spatial gradients, although it does not update parameters at every time step. Excluding the regularization terms STGR and MPDR we introduced, the base loss function across different datasets is identical to that of SLTT. Note that the theoretical formulation of the Spatio-Temporal Gradient Regularization (STGR) includes a gradient norm penalty term, i.e., $\frac{\gamma}{2}\|\nabla l_{t-1}(W_t)\|^2$, as defined in Eq. 5. Exact differentiation of this term with respect to $W_t$ would technically require second-order information (Hessian-vector products), which typically incurs a memory complexity of $O(N^2)$ or significant computational overhead. To strictly maintain the $O(N)$ memory complexity required for efficient online learning, we implement this term using a first-order approximation strategy.

**Hyperparameters.** For all the tasks, we use SGD (Rumelhart et al. (1986)) with momentum 0.9 to train the networks, and use cosine annealing (Loshchilov & Hutter (2016)) as the learning rate schedule. For both OSBP and RPTT, in order to accurately evaluate the effect of our designed regularization terms, we adopt exactly the same hyperparameter configurations across all tasks, and only adjust the regularization terms specifically for RPTT. For CIFAR-10 and CIFAR-100, the number of epochs, learning rate(LR), batch size(BS), and weight decay(WD) are set to 200, 0.1, 128, and $5 \times 10^{-5}$, respectively. For ImageNet, we employ an SLTT pre-trained model and perform fine-tuning using RPTT only, with the fine-tuning process configured as 30 epochs, LR of 0.001, BS of

256, and WD of 0. For DVS-CIFAR10, the settings are 300 epochs, LR of 0.05, BS of 128, and WD of $5 \times 10^{-4}$. The number of time steps in the four experiments is set to $T = 6, 6, 6$, and 10, respectively. To simplify hyperparameter tuning, we discard the fine-grained modeling of the target distribution and hyperparameters at different membrane potential layers (which in practice could yield better results), and instead adopt a unified setting for all layers. Accordingly, we apply a relatively strong constraint: the target distribution of MPDR is set to $q = \mathcal{N}(-60, 10)$. For all datasets, we fix the coefficient of the second term in STGR to $\gamma_{STGR} = 0.5$ and the variance-term coefficient in MPDR to $\mu_{MPDR} = 0.01$. We then adjust the regularization strengths $\lambda_{STGR}$ and $\lambda_{MPDR}$ according to the dataset scale. Specifically, for CIFAR-10, we set $\lambda_{STGR} = 0.006$ and $\lambda_{MPDR} = 0.002$; for CIFAR-100 and DVS-CIFAR10, we set $\lambda_{STGR} = 0.009$ and $\lambda_{MPDR} = 0.0035$; and for ImageNet-1K, we set $\lambda_{STGR} = 0.003$ and $\lambda_{MPDR} = 0.0006$. The $\rho$ of Equation 6 decays with training epochs, which is $\rho = \max\{0.01, 1/(1 + 0.1 \cdot epoch)\}$.

### A.2.5 ADDITIONAL VISUALIZATION EXPERIMENTS

In this section, we present supplementary visualizations to offer a comprehensive analysis of the membrane potential distribution across different layers in RPTT. While the main text highlights the bimodal distribution in the first layer of ResNet-18 trained by RPTT, our findings indicate that from the second layer onward, the distribution closely approximates a standard Gaussian. This phenomenon can be attributed to the escalating complexity of neuronal dynamics in SNNs with increasing network depth. The initial bimodal distribution is progressively smoothed through cumulative transformations across subsequent layers, leading to a gradual transition from a bimodal to a unimodal Gaussian distribution. We conduct experiments on CIFAR-10. Under identical settings, we feed the same samples into a ResNet-18 trained with BPTT, OSBP, and RPTT. We then visualize the membrane potential distribution of the fifteenth layer over six time steps (Figure 5, 6, 7). We also quantified the degree of membrane potential drift from another perspective: the proportion of neurons whose membrane potential exceeds the firing threshold at the same time step on three methods (Figure 8).

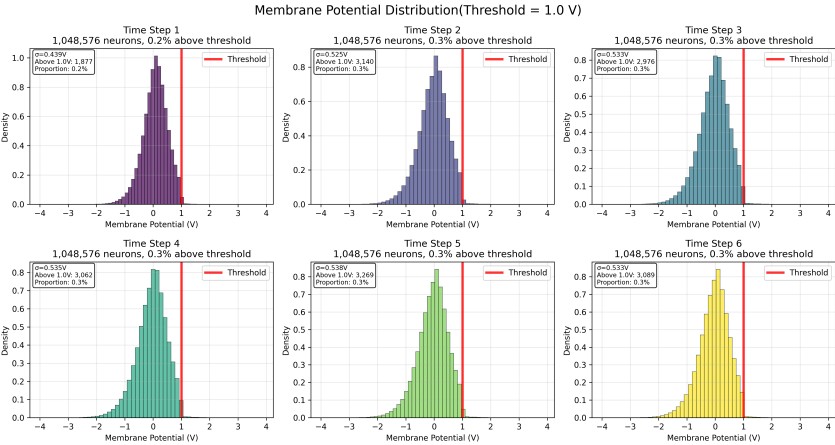

Figure 5: For the BPTT-trained ResNet-18 network on the CIFAR-10 dataset, we visualize the membrane potential distributions of neurons at the fifteenth layer across time steps 1 to 6, with the threshold potential denoted as $v_t h$ and the y-axis representing the probability density. All subsequent distributions adhere to these settings.

As shown in Figure 5, 6, 7, in deeper layers, BPTT maintains a Gaussian distribution, indicating a sustained level of neuronal activity. Conversely, the distribution for OSBP becomes significantly narrower, implying that a large proportion of neurons enter a quiescent state, thereby diminishing the network's representational capacity. This conclusion is further corroborated by the percentage of neurons exceeding the firing threshold. In the fifteenth layer, approximately 0.3% of neurons in the BPTT-trained model surpass the threshold, while this figure is nearly zero for the OSBP trained model. In contrast, RPTT, leveraging the MPDR mechanism, successfully stabilizes the membrane potential distribution. It restores the narrow distribution seen in OSBP to a Gaussian form with greater variance. This indicates that our method effectively restores spiking activity that is essen-

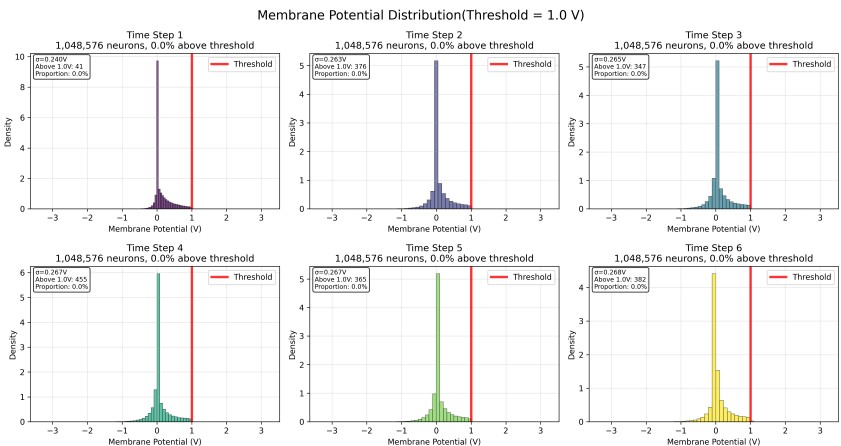

Figure 6: Neuron membrane potential distribution under OSBP training

tial for learning. Consistently, about 0.1% of neurons in the RPTT model exceed the threshold, demonstrating that RPTT enhances the expressive power of neurons and mitigates membrane potential drift. Figure 8 shows the proportion of neurons exceeding the threshold across all layers of ResNet-18 for the first time step using BPTT, OSBP, and RPTT. The results indicate that the degree of membrane potential distribution drift in RPTT is consistently lower than that in OSBP across all layers, which can be attributed to the layer-wise constraint imposed by MPDR.

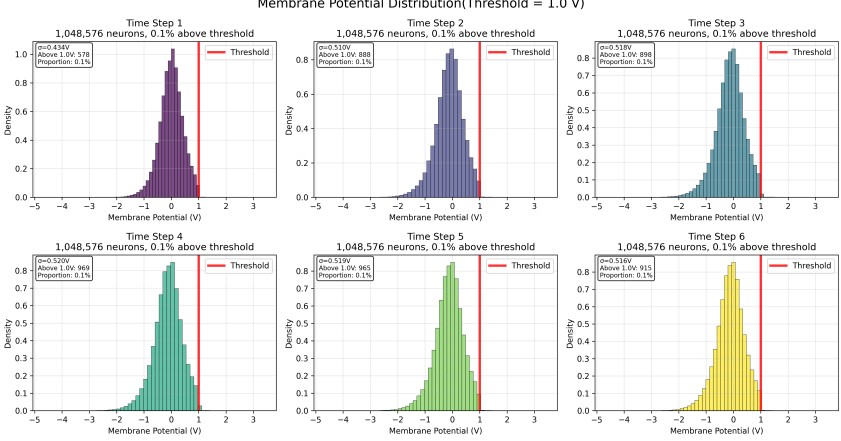

Figure 7: Neuron membrane potential distribution under RPTT training

In addition, to empirically validate the stability gains conferred by STGR, we conducted a comparative analysis on the CIFAR-10 dataset. This experiment evaluated two settings: the baseline OSBP algorithm and OSBP integrated with our proposed STGR. To ensure a rigorous comparison, both settings utilized identical hyperparameters and were executed over five independent runs, each with a distinct random seed. OSBP with STGR achieves a smaller variance (0.116%) compared to OSBP (0.149%). The results demonstrate that the integration of STGR enhances training stability.

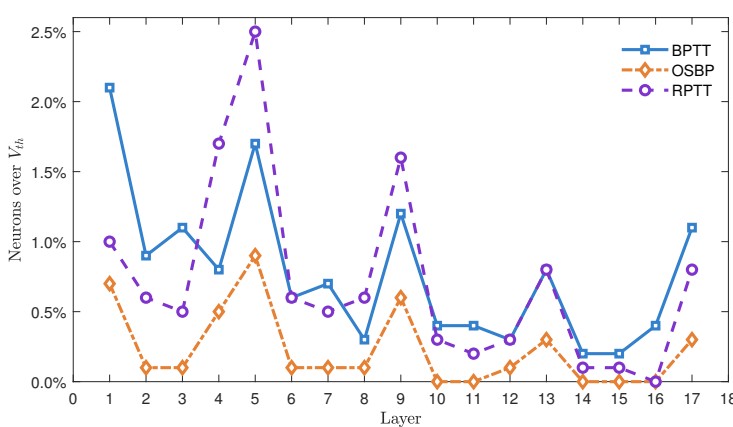

Figure 8: The proportion of neurons in different layers exceeding the voltage threshold

REFERENCES FOR APPENDIX

Andrew Brock, Soham De, and Samuel L Smith. Characterizing signal propagation to close the performance gap in unnormalized resnets. *arXiv preprint arXiv:2101.08692*, 2021a.

Andy Brock, Soham De, Samuel L Smith, and Karen Simonyan. High-performance large-scale image recognition without normalization. In *International conference on machine learning*, pp. 1059–1071. PMLR, 2021b.

Jia Deng, Wei Dong, Richard Socher, Li-Jia Li, Kai Li, and Li Fei-Fei. Imagenet: A large-scale hierarchical image database. In *2009 IEEE conference on computer vision and pattern recognition*, pp. 248–255. Ieee, 2009.

Shikuang Deng, Yuhang Li, Shanghang Zhang, and Shi Gu. Temporal efficient training of spiking neural network via gradient re-weighting. *arXiv preprint arXiv:2202.11946*, 2022.

Wei Fang, Yanqi Chen, Jianhao Ding, Zhaofei Yu, Timothée Masquelier, Ding Chen, Liwei Huang, Huihui Zhou, Guoqi Li, and Yonghong Tian. Spikingjelly: An open-source machine learning infrastructure platform for spike-based intelligence. *Science Advances*, 9(40):eadi1480, 2023.

Kaiming He, Xiangyu Zhang, Shaoqing Ren, and Jian Sun. Deep residual learning for image recognition. In *Proceedings of the IEEE conference on computer vision and pattern recognition*, pp. 770–778, 2016a.

Kaiming He, Xiangyu Zhang, Shaoqing Ren, and Jian Sun. Identity mappings in deep residual networks. In *European conference on computer vision*, pp. 630–645. Springer, 2016b.

Haiyan Jiang, Giulia De Masi, Huan Xiong, and Bin Gu. Ndot: Neuronal dynamics-based online training for spiking neural networks. In *Forty-first International Conference on Machine Learning*, 2024.

Alex Krizhevsky, Geoffrey Hinton, et al. Learning multiple layers of features from tiny images. 2009.

Hongmin Li, Hanchao Liu, Xiangyang Ji, Guoqi Li, and Luping Shi. Cifar10-dvs: an event-stream dataset for object classification. *Frontiers in neuroscience*, 11:244131, 2017.

Yuhang Li, Yufei Guo, Shanghang Zhang, Shikuang Deng, Yongqing Hai, and Shi Gu. Differentiable spike: Rethinking gradient-descent for training spiking neural networks. *Advances in neural information processing systems*, 34:23426–23439, 2021.

Ilya Loshchilov and Frank Hutter. Sgdr: Stochastic gradient descent with warm restarts. *arXiv preprint arXiv:1608.03983*, 2016.

Qingyan Meng, Mingqing Xiao, Shen Yan, Yisen Wang, Zhouchen Lin, and Zhi-Quan Luo. Training high-performance low-latency spiking neural networks by differentiation on spike representation. In *Proceedings of the IEEE/CVF conference on computer vision and pattern recognition*, pp. 12444–12453, 2022.

Qingyan Meng, Mingqing Xiao, Shen Yan, Yisen Wang, Zhouchen Lin, and Zhi-Quan Luo. Towards memory-and time-efficient backpropagation for training spiking neural networks. In *Proceedings of the IEEE/CVF international conference on computer vision*, pp. 6166–6176, 2023.

Adam Paszke, Sam Gross, Francisco Massa, Adam Lerer, James Bradbury, Gregory Chanan, Trevor Killeen, Zeming Lin, Natalia Gimelshein, Luca Antiga, et al. Pytorch: An imperative style, high-performance deep learning library. *Advances in neural information processing systems*, 32, 2019.

Siyuan Qiao, Huiyu Wang, Chenxi Liu, Wei Shen, and Alan Yuille. Micro-batch training with batch-channel normalization and weight standardization. *arXiv preprint arXiv:1903.10520*, 2019.

David E Rumelhart, Geoffrey E Hinton, and Ronald J Williams. Learning representations by back-propagating errors. *nature*, 323(6088):533–536, 1986.

Karen Simonyan and Andrew Zisserman. Very deep convolutional networks for large-scale image recognition. *arXiv preprint arXiv:1409.1556*, 2014.

Nitish Srivastava, Geoffrey Hinton, Alex Krizhevsky, Ilya Sutskever, and Ruslan Salakhutdinov. Dropout: a simple way to prevent neural networks from overfitting. *The journal of machine learning research*, 15(1):1929–1958, 2014.

Mingqing Xiao, Qingyan Meng, Zongpeng Zhang, Di He, and Zhouchen Lin. Online training through time for spiking neural networks. *Advances in neural information processing systems*, 35:20717–20730, 2022.

Hanle Zheng, Yujie Wu, Lei Deng, Yifan Hu, and Guoqi Li. Going deeper with directly-trained larger spiking neural networks. In *Proceedings of the AAAI conference on artificial intelligence*, volume 35, pp. 11062–11070, 2021.

