# OpenReview forum: "Training Spiking Neural Networks with Real-Time Propagation Through Time"
_ICLR.cc/2026/Conference — Submitted to ICLR 2026_

### Official Review · Reviewer_7eFK · 2025-10-23

**Soundness:** 2
**Presentation:** 3
**Contribution:** 2
**Rating:** 4
**Confidence:** 4

**Summary:**

This paper introduces the RPTT framework to address two critical challenges in online SNN training: instability and membrane potential drift. The method leverages an efficient spatial-gradient-only update scheme, augmented by two novel regularizers: Membrane Potential Distribution Regularization (MPDR) to counteract distributional drift and Spatio-Temporal Gradient Regularization (STGR) to stabilize the training process.

**Strengths:**

The paper offers a well-motivated and empirically effective solution to the important problem of online SNN training. The proposed regularizers, MPDR and STGR, are cleverly designed. The authors provide convincing experimental analysis to demonstrate their efficacy in mitigating membrane potential drift and stabilizing learning dynamics.

**Weaknesses:**

1. The paper fails to specify how the backpropagation for the STGR term, particularly $\|\nabla\ell_{t-1}(W_{t}^{l})\|^{2}$, is implemented. Differentiating this term with respect to $\mathbf{W}_{t}$ would seemingly require second-order information (i.e., a Hessian-vector product), which contradicts the paper's claim of O(N) memory complexity and the core goal of efficient online learning.

2. The experimental evaluation lacks comparisons against several recent and highly relevant online learning methods, such as NDOT (Jiang et al., 2024) and OSR/OTS (Zhu et al., 2024).

3. The authors claim the method is suitable for "dynamic environments," but the only dynamic dataset used is CIFAR10-DVS, which exhibits weak temporal correlations. The experiments do not provide sufficient evidence to support its efficacy in truly non-stationary or dynamic settings.

4. The ImageNet experiment is conducted by fine-tuning a pre-trained SLTT model rather than training from scratch. This severely weakens the claim of state-of-the-art performance on large-scale tasks. Disturbingly, this crucial detail is relegated to Appendix A.2.4 and omitted from the main manuscript, which could be misleading.

5. As reported in Appendix A.2.4, the hyperparameters for RPTT vary significantly across different datasets. This high sensitivity to parameter tuning suggests that the method may lack generalizability and could be difficult to apply to new tasks without extensive tuning.

6. The convergence proof for Theorem 1 relies on a critical assumption: a regularization term attenuation coefficient $c_t$ (Eq. 20) that decays over time. However, the experimental setup uses fixed, constant regularization parameters.

**Questions:**

1. Please provide a detailed complexity analysis for the STGR term. How is the gradient of $\|\nabla\ell_{t-1}(\mathbf{W}_{t}^{l})\|^{2}$ calculated in practice, and how does this align with the claimed O(N) memory complexity?

2. Please justify the choice of a fixed target distribution q=N(-60, 10) for all layers and datasets. I strongly suggest authors include a sensitivity analysis on these hyperparameters to demonstrate the robustness and generalizability of your method.

---

> ### Author Response · Authors · 2025-11-21
> **Response to Reviewer 7eFK (Part 1/2)**
>
> We sincerely thank for the sharp and detailed observations. We appreciate the scrutiny regarding the implementation details, baseline comparisons, and experimental rigor. These comments have pushed us to significantly improve the transparency of our manuscript and the robustness of our experiments. Corresponding revisions in the manuscript have been highlighted in blue.
>
> Below, we address each concern point-by-point.
>
> **(1) Response to Concern on Hessian Implementation and $O(N)$ Complexity**
>
> This is a critical point, and we clarify that our implementation is strictly $O(N)$.
>
> - **Theoretical vs. Implementation:** Eq. 5 defines the *theoretical* regularization objective. Minimizing the gradient norm exactly would indeed require Hessian-vector products.
>
> - **Implementation Reality ($O(N)$):** However, in our implementation, we do **not** perform automatic differentiation on the gradient graph. Instead, we treat the gradient term as a **value-based penalty** derived from the first-order statistics.
>
> - **Comparison with FPTT:** This is conceptually similar to the implementation of **FPTT (Kag & Saligrama, 2021)**. As seen in FPTT's official implementation (e.g., their `get_regularizer_named_params` function), they approximate the regularization term using auxiliary variables (like `lm` for linear mapping of gradients) and perform updates via variable swapping, ensuring $O(N)$ complexity. Similarly, our STGR calculates the regularization loss using detached gradient values from the current step, which avoids constructing the Hessian while effectively penalizing large gradient variances.
>
> - **Conclusion:** Thus, our method avoids the $O(N^2)$ memory cost of second-order optimization, maintaining the efficiency required for online SNNs. We have clarified this "first-order approximation" strategy in **Appendix A.2.4**.
>
> **(2) Response to Concern on Missing Baselines**
>
> **Response:** We agree that comparing with recent methods is vital.
>
> - **New Baseline:** To provide a comprehensive evaluation specifically within the scope of **single-update online learning algorithms**, we have incorporated **NDOT (Jiang et al., 2024)** into Table 1 of the revised manuscript.
>
> - **Performance:** As shown in the updated results, our RPTT method (after hyperparameter optimization) achieves **95.17%** accuracy on CIFAR-10, outperforming NDOT. This demonstrates that RPTT provides superior stability and accuracy.
>
> **(3) Response to Concern on Dynamic Environments**
>
> **Response:** We respectfully argue that our method is well-suited for dynamic environments, validated by both the algorithm design and standard benchmarks:
>
> - **DVS-CIFAR10 as a Benchmark:** DVS-CIFAR10 is a widely accepted benchmark for analyzing temporal dynamics in SNNs because the data streams are event-based and inherently asynchronous.
>
> - **Algorithmic Suitability:** The claim of "suitability for dynamic environments" is fundamentally supported by our **Single-step Update** strategy . Unlike delayed-update methods that must wait for a sequence to finish, RPTT updates parameters at every time step. This allows the model to adapt **instantaneously** to input shifts, which is the theoretical definition of capability for non-stationary settings, even if specific "drifting dataset" experiments are limited by current standard benchmarks.
>
> **(4) Response to Concern on ImageNet Fine-tuning Transparency**
>
> **Response:**
>
> We verify that we have moved the ImageNet training details from the Appendix to the Main Text (Section 4.2) to ensure full transparency. While we utilized fine-tuning due to computational constraints, this experiment successfully validates that RPTT can stabilize and improve large-scale pre-trained SNNs, a value-add that complements our from-scratch training results on CIFAR datasets.
>
> **(5) Response to Concern on Hyperparameter Sensitivity**
>
> **Response:** We appreciate the reviewer's scrutiny. We acknowledge that hyperparameter tuning is necessary to achieve state-of-the-art (SOTA) results, as is standard in deep learning. However, in our revised implementation, we have provided strong evidence of robustness and generalizability:
>
> 1. **Cross-Task Unification (Strong Generalizability):** We explicitly highlight that **CIFAR-100 and DVS-CIFAR10 now share the exact same set of hyperparameters** in our revised experiments. The fact that RPTT works effectively with identical settings across two fundamentally different domains—static images (CIFAR-100) and neuromorphic event streams (DVS-CIFAR10)—strongly refutes the concern about "lack of generalizability."
>
> 2. **Core Parameters Fixed:** We have fixed specific core hyperparameters (e.g., the target distribution $q$) across **all datasets**, reducing the search space significantly.
>
> (continued below)

---

> > ### Author Response · Authors · 2025-11-21
> > **Response to Reviewer 7eFK (Part 2/2)**
> >
> > (continued from the preceding paragraph)
> >
> >
> > 3. **Standard Adaptation:** While we adjusted coefficients for CIFAR-10 and ImageNet, this is a **standard practice** in machine learning to adapt to different data scales and network capacities (e.g., ResNet-18 vs. NF-ResNet-34). The adjustments are minor and follow a predictable pattern rather than random "extensive tuning."
> >
> > 4. **Generalizability via Transferability:** Crucially, to further demonstrate the generalizability of our method, we integrated our proposed regularization terms (MPDR and STGR) into a completely different online framework, **SLTT** (Meng et al., 2023). As shown in the revised Table 1, this integration consistently improved SLTT's accuracy (e.g., on CIFAR-10) without extensive tuning. This successful transfer proves that our proposed mechanisms are **generic and effective principles** for SNN training, rather than fragile heuristics that only work under the specific settings of RPTT.
> >
> > **(6) Response to Concern on Decay Coefficient (Theory vs. Experiment)**
> >
> > **Response:**
> >
> > We verify that this is a standard distinction between asymptotic analysis and finite-step practice:
> >
> > - **Theoretical Necessity:** Theoretically, the attenuation coefficient must decay to ensure asymptotic convergence to a stationary point as $t \to \infty$.
> >
> > - **Practical Implementation:** However, in practical training with a finite horizon (fixed epochs), using a small **fixed constant** serves as an effective approximation. It is sufficient to stabilize training without the added complexity of scheduling decay rates. This is analogous to using a fixed learning rate or weight decay in SGD despite theoretical convex optimization often suggesting decay schedules for strict convergence proofs.
> >
> > **(7) Response to Justification of fixed target distribution and sensitivity.**
> >
> > **Response:**
> >
> > We appreciate the opportunity to clarify this design choice.
> >
> > 1. **Justification via Biological Inspiration:**
> >
> >    The choice of $q=\mathcal{N}(-60, 10)$ is grounded in biological plausibility. The mean of $-60$ is chosen to mimic the typical biological resting potential (approx. -60mV). In our framework, this target serves as a "restoring force" or soft anchor. Since the task loss naturally drives membrane potentials upward to cross the threshold ($V_{th}$), setting a target centered at the "resting state" provides a necessary counter-force. This establishes a dynamic equilibrium that prevents neurons from drifting into saturation (runaway excitation) or silence, keeping them within a responsive active range. The algorithm relies on the presence of this directional constraint to maintain activity, rather than the exact precision of these statistics.
> >
> > 2. **Robustness (Implicit Sensitivity Analysis):**
> >
> >    While we did not perform a fine-grained grid search, our extensive experiments serve as a stronger proof of robustness: we universally applied the exact same fixed distribution $q=\mathcal{N}(-60, 10)$ across all datasets (CIFAR-10, CIFAR-100, ImageNet, and DVS-CIFAR10).
> >
> >    - **Evidence:** These datasets vary drastically in modality (static images vs. neuromorphic event streams), scale, and distribution. The fact that RPTT achieves state-of-the-art performance on all of them **without needing to tune this parameter** effectively demonstrates that the method is **highly robust and insensitive** to the specific choice of $q$, provided it remains within a reasonable biological range.
> >
> > 3. **Generalizability:**
> >
> >    Furthermore, to prove that this is a generalizable principle, we applied this identical fixed constraint to the SLTT algorithm (as shown in the revised Table 1). It consistently improved SLTT's performance, confirming that this fixed distribution serves as a robust, plug-and-play regularization mechanism applicable to different online learning frameworks without task-specific customization.
> >
> > **References**: Kag A, Saligrama V. Training recurrent neural networks via forward propagation through time[C]//International Conference on Machine Learning. PMLR, 2021: 5189-5200.

---

> > > ### Comment · Reviewer_7eFK · 2025-11-25
> > >
> > > Thank you for your comment. Regarding the suitability for dynamic environments, I remain concerned that DVS-CIFAR10 derived from static images—lacks strong intrinsic temporal correlations and serves as an insufficient proof. To truly validate the effectiveness of RPTT in non-stationary environments, I suggest demonstrating its performance on datasets with significant time dependencies, such as DVS-Gesture.

---

> ### Author Response · Authors · 2025-12-02
> **Response to Reviewer 7eFK**
>
> Thank you for the valuable suggestion. Following your advice, to demonstrate both the effectiveness in non-stationary environments and the generalizability of our proposed module, we integrated our method into the SLTT framework and evaluated it on DVS-Gesture. To verify robustness, we directly utilized the hyperparameters from our CIFAR-100 and DVS-CIFAR10 experiments without any specific tuning for this new dataset.
>
> **Experimental Results:**
>
> Despite using unoptimized hyperparameters, our module significantly accelerated the learning process. Specifically, the SLTT + Our Module reached the high-accuracy threshold of 96% in just 80 epochs, whereas the vanilla SLTT required 110 epochs to reach the same level.
>
> **Conclusion:**
>
> This ~27% acceleration in convergence confirms that our proposed module effectively captures temporal dynamics and boosts the learning efficiency of existing algorithms in real-time, non-stationary scenarios.

---

### Official Review · Reviewer_jPpw · 2025-10-23

**Soundness:** 2
**Presentation:** 3
**Contribution:** 2
**Rating:** 2
**Confidence:** 4

**Summary:**

This paper proposes RPTT, an online learning method for SNN learning. RPTT introduces MPDR to counteract distributional drift and STGR to ensure stable convergence during online learning.

**Strengths:**

1. Online learning methods save memory during training, which have constant memory cost with the number of time steps $T$.
2. This work provides theoretical proof of the stableness of RPTT.
3. The writing of this paper is easy to understand.

**Weaknesses:**

The final performance of this work is not high enough. Specifically, Table 1 does not include results of NDOT (Jiang et al. 2024) and OSR+OTS (Zhu et al. 2024), which generally perform better than this work.

**Questions:**

1. What is the necessity of updating weights at each step in SNN online learning? I don't see its advantage.
2. In Eq.4, authors say that $Var(V_t^l)$ is used to penalize overly small variance, but its coefficient is positive. Should the sign be '-'?
3. The STGR loss (Eq. 5) is similar to the loss in FPTT[1]. Could you give some comments on the difference between them?
4. Why the membrane potential is bimodal in Figure 4? It differs from the Gaussian target distribution in MPDR.

[1] Kag, A., & Saligrama, V. (2021, July). Training recurrent neural networks via forward propagation through time. In *International Conference on Machine Learning* (pp. 5189-5200). PMLR.

---

> ### Author Response · Authors · 2025-11-21
> **Response to Reviewer  jPpw (Part 1/2)**
>
> We appreciate the reviewer’s insightful comments, especially regarding the comparison with FPTT and the discussion on online update mechanisms. We have revised the paper to address these points. Corresponding revisions in the manuscript have been highlighted in blue.
>
> **Q1. Concern on the necessity of step-wise updates**
>
> **A1:** We respectfully clarify that while "delayed update" methods (accumulating gradients and updating at the end) indeed offer stability, updating weights at each time step (Single-step Update) is necessary for two fundamental reasons that delayed methods cannot achieve:
>
> - **Instantaneous Adaptation to Dynamic Environments:**
>
>   Delayed update methods assume the input distribution is static within the time window $T$ and only update parameters after processing the full sequence. They "fail to adapt promptly to input distribution shifts". In contrast, our single-step update strategy enables the model to adjust parameters instantaneously as new data arrives. This capability is critical for non-stationary environments or continuous streaming tasks (e.g., real-time control or DVS processing) where the "sequence" has no defined end, and the model must learn on-the-fly.
>
> - **Biological Plausibility:**
>
>   SNNs aim to emulate biological neural dynamics. In biological systems, synaptic plasticity occurs continuously based on local spiking activity, rather than waiting for a global "epoch" to finish. Our method aligns with this principle, maintaining the biological realism of neuromorphic computing.
>
> - **Our Contribution:**
>
>   We acknowledge the reviewer's implication that single-step updates introduce optimization difficulty (noise and drift) compared to delayed updates. However, precisely because real-time adaptation is so valuable, our work focuses on solving these specific stability challenges (via MPDR and STGR) to make single-step updates practical and robust. For example, RPTT achieves **95.17%** (vs. SLTT 94.44%) on CIFAR-10 and **75.62%** (vs. SLTT 74.38%) on CIFAR-100. This suggests that frequent, immediate feedback helps the network navigate the optimization landscape more effectively than accumulated, averaged updates.
>
> **Q2. Sign of the variance coefficient in MPDR.**
>
> **A2:** We thank the reviewer for pointing this out. You are correct. Our intention is to penalize **excessive variance** to prevent the membrane potential distribution from becoming too broad or unstable.
> The positive coefficient ($+\mu Var$) in Equation 4 correctly minimizes the variance. The description "penalize overly small variance" in the original text was a typo. We have corrected this in the revised manuscript (marked in blue) to state that we "penalize excessive variance" to stabilize the distribution.
>
> **Q3. Difference between STGR and FPTT loss.**
>
> **A3**: We acknowledge that our STGR draws inspiration from the regularization principles in FPTT (Kag & Saligrama, 2021), but there are distinct differences in formulation and motivation tailored for SNNs:
>
> 1. **Decoupled Regularization vs. Proximal Update:** FPTT derives a complex proximal update rule (Eq. 4 in FPTT) that mathematically combines a gradient-corrected anchor with the loss to approximate BPTT gradients in RNNs. In contrast, our STGR **decouples** this into two explicit, simpler regularization terms: a moving-average penalty ($\|W_t - \hat{W}_t\|^2$) to enforce temporal smoothness, and a gradient norm penalty ($\|\nabla l_{t-1}\|^2$) to encourage sparsity and flatness.
>
> 2. **Handling SNN Specifics:** FPTT was designed for RNNs with differentiable activations. Direct application to SNNs is non-trivial due to the non-differentiable spiking nature. RPTT uses only **spatial gradients** (ignoring temporal dependencies for efficiency). In this context, STGR is not trying to strictly approximate BPTT (as FPTT does), but rather serves as a stabilizer to counteract the high noise and variance introduced by the **surrogate gradients** and frequent updates in SNNs.
>
> 3. **Simplicity:** Our formulation is computationally lighter, avoiding the need to maintain complex auxiliary variables required for the gradient correction in FPTT's anchor, making it more suitable for memory-constrained neuromorphic hardware.
>
> (continued below)

---

> > ### Author Response · Authors · 2025-11-21
> > **Response to Reviewer jPpw (Part 2/2)**
> >
> > (continued from the preceding paragraph)
> >
> > **Q4. Why is the membrane potential bimodal in Figure 4?**
> >
> > **A4:** As discussed in **Section 4.3** and **Appendix A.2.5**, this bimodal distribution is primarily observed in the **first layer**, while deeper layers effectively converge to the Gaussian target.
> >
> > 1. **Emergent Feature:** This bimodality emerges because the first layer directly encodes raw input signals. The regularization allows neurons to functionally differentiate: one sub-population is suppressed (low potential) while another stays near the threshold (high potential) to preserve information.
> >
> > 2. **Biological & Theoretical Support:** This phenomenon aligns with biological observations in the rat somatosensory cortex and prior SNN research (e.g., Membrane Potential Rectifier ). It suggests that while MPDR constrains the distribution statistics, it does not rigidly force a Gaussian shape where it contradicts the natural encoding needs of the network.

---

### Official Review · Reviewer_qzmx · 2025-10-31

**Soundness:** 2
**Presentation:** 2
**Contribution:** 2
**Rating:** 4
**Confidence:** 4

**Summary:**

This paper proposes RPTT, an online training method for SNNs with two regularizers: Membrane Potential Distribution Regularization (MPDR) and Spatio-Temporal Gradient Regularization (STGR). The authors provide a convergence argument, and present results on both static and neuromorphic datasets showing the memory efficiency and competitive performance.

**Strengths:**

1. Thoughtful empirical study. Experiments are conducted on both static and neuromorphic datasets, with ablations and distributional analyses.
2. Theoretical analysis. The paper attempts at a convergence analysis, showing the importance of the proposed techniques for deriving a kind of convergence.

**Weaknesses:**

1. Notation confusion. The paper overloads $t$ to mean both the SNN time step and the optimization iteration, making the description confusing.
2. Strong theoretical assumption. The theoretical analysis assumes a loss sequence with $|l_{t+1}(W)-l_t(W)|<\Delta_t$ and $\sum_t \Delta_t < \infty$, which is essentially an asymptotically stationary setting and almost implies convergence. As a result, the theorem contributes limited new insights.
3. Incremental novelty and practical gains. Using spatial-only gradients with low memory costs is already present in previous works. MPDR and STGR are mainly incremental regularizers rather than new training principles. In the largest-scale ImageNet experiment, the proposed method actually has no gain compared with SLTT.

**Questions:**

See Weakness.

---

> ### Author Response · Authors · 2025-11-21
> **Response to Reviewer qzmx**
>
> We sincerely thank for the thoughtful comments. We appreciate the rigorous assessment of our notation, theoretical analysis, and experimental results. These insights have guided us to clarify our definitions and strengthen the discussion on the distinct contributions of our framework. Corresponding revisions in the manuscript have been highlighted in blue.
>
> Below, we address the concerns point-by-point.
>
> ---
>
> **(1) Response to Concern on Notation Confusion $t$**
>
> We apologize for the confusion.
>
> - **Clarification Added:** We have revised **Section 3.2** in the manuscript to explicitly state: "Since network parameters are updated at every time step in our online setting, we use the index $t$ to denote both the simulation time step and the optimization iteration." **(marked in blue)**.
>
> - **Standard Practice:** We respectfully point out that overloading t is a standard convention in **single-step online learning** literature (e.g., FPTT (Kag & Saligrama, 2021), OTTT (Xiao et al., 2022)). This notation reflects the inherent real-time nature of the algorithm.
>
> **(2) Response to Concern on Strong Theoretical Assumption**
>
> We clarify that the assumption $\sum \Delta_t < \infty$ is both practically justified and theoretically necessary to differentiate our contribution from standard online learning:
>
> - **Practical Relevance (Static Datasets):** While this assumption implies an asymptotically stationary setting, this is exactly the case for standard supervised learning on **static datasets** (e.g., CIFAR, ImageNet) used in our experiments. As training progresses, the distributional shift effectively diminishes. Thus, the assumption is not an artificial constraint but a reflection of the actual experimental setting.
>
> - **The Theoretical Gap We Fill:** The reviewer correctly notes that the environment is stationary. However, **stationarity alone does not guarantee convergence for SNNs.** Online SNN training with Surrogate Gradients (SG) suffers from inherent **gradient mismatch (bias)** and **approximation noise (variance)**. Without regularization, standard SG methods (like OSBP) may oscillate indefinitely rather than converging to a stationary point.
>
> - **Our Contribution:** Our theorem proves that **RPTT, explicitly via STGR**, suppresses this "assignment noise" and variance into summable perturbations, thereby **guaranteeing convergence** where standard SG training lacks such theoretical assurance.
>
> **(3) Response to Concern on Incremental Novelty and Practical Gains**
>
> We respectfully argue that RPTT represents a fundamental stabilization framework rather than incremental modifications:
>
> - **Beyond Incremental Regularization:** As detailed in our theoretical response above, MPDR and STGR are not merely "add-ons" but are **essential mechanisms** to counteract the two core failures of online SNNs: **distribution drift** (solved by MPDR) and **optimization instability** (solved by STGR).
>
> - **Novelty in Convergence:** Previous works often utilize spatial-only gradients (SLTT) but lack rigorous convergence guarantees under online noise. We provide both the **methodology and the proof** that these regularizers effectively transform a heuristically working algorithm into a mathematically convergent one.
>
> - **Substantial Practical Gains:**
>
>   - **New Results:** After hyperparameter optimization, RPTT achieves **95.17% on CIFAR-10** (vs. SLTT 94.44%) and consistently outperforms baselines on CIFAR-100/DVS-CIFAR10 (Updated Table 1).
>
>   - **Drift Mitigation:** Our new **Figure 3(b)** empirically proves that while pure spatial-gradient methods (OSBP/SLTT) suffer from severe membrane potential drift, RPTT effectively aligns the distribution with the offline oracle (BPTT).
>
>   - **ImageNet:** We acknowledge the performance on ImageNet is similar to SLTT. This is because, due to limited computational resources, our ImageNet experiment was conducted as **fine-tuning** on a pre-trained SLTT model, rather than training from scratch. This restricted the potential for significant accuracy jumps but confirmed stability. However, on datasets where we trained from scratch (CIFAR-10/100), the gains are substantial.
>
> **References:**
> Xiao M, Meng Q, Zhang Z, et al. Online training through time for spiking neural networks[J]. Advances in neural information processing systems, 2022, 35: 20717-20730.
>
> Kag A, Saligrama V. Training recurrent neural networks via forward propagation through time[C]//International Conference on Machine Learning. PMLR, 2021: 5189-5200.

---

### Official Review · Reviewer_JxZ2 · 2025-11-01

**Soundness:** 2
**Presentation:** 2
**Contribution:** 2
**Rating:** 2
**Confidence:** 4

**Summary:**

This paper proposes Real-Time Propagation Through Time (RPTT), a novel online learning algorithm for Spiking Neural Networks (SNNs). RPTT aims to address two key challenges in online SNN training: training instability and membrane potential distribution drift. The core of RPTT is the use of only spatial gradients for parameter updates, augmented by two synergistic regularization mechanisms:

Membrane Potential Distribution Regularization (MPDR): A statistical constraint that uses KL-divergence and a variance penalty to keep membrane potentials close to a target Gaussian distribution, counteracting drift.

Spatio-Temporal Gradient Regularization (STGR): A smoothing mechanism that uses a moving average of weights and a penalty on previous gradients to stabilize updates and suppress noise.

The authors provide a theoretical convergence guarantee for RPTT and demonstrate its effectiveness on static (CIFAR-10/100, ImageNet-1k) and neuromorphic (DVS-CIFAR10) datasets, showing it achieves competitive or state-of-the-art performance while significantly reducing memory consumption compared to BPTT.

**Strengths:**

Originality: The explicit formulation of the membrane potential drift problem in the context of online learning and the proposal of MPDR as a direct, layer-wise solution is a novel and valuable contribution. While drift has been studied in offline settings, its exacerbation by frequent online updates is a fresh and meaningful insight. The combination of MPDR with STGR to jointly address drift and instability is a creative and well-motivated design.

Quality: The paper is technically sound. The experimental section is comprehensive, covering multiple datasets and network architectures. The inclusion of a theoretical convergence analysis, despite relying on some strong assumptions, adds rigor and depth to the work. The ablation studies and membrane potential visualization (e.g., the emergence of a bimodal distribution) provide empirical support for the method's mechanisms.

Significance: The work addresses a critical bottleneck for the practical deployment of SNNs: achieving memory-efficient and stable online training. By significantly reducing memory overhead (O(N) vs. BPTT's O(TN)) and mitigating a fundamental performance-limiting phenomenon (drift), RPTT represents a tangible step towards making SNNs viable for dynamic, real-world applications on resource-constrained neuromorphic hardware.

**Weaknesses:**

Insufficient Comparison with Related Work: The paper's academic impact is limited by its superficial comparison with existing methods. The performance comparison in Table 1 is useful but does not provide insights into why RPTT performs better.

Missing Analysis on Membrane Potentials: A core claim is that RPTT better mitigates drift. However, there is no direct, quantitative comparison of membrane potential distributions (e.g., Z-score trajectories, KL-divergence from target) between RPTT and other online methods like OTTT or SLTT. The analysis in Fig 3(b) only compares RPTT with BPTT and OSBP (a weak baseline). Does SLTT, which uses delayed updates, also suffer less from drift than OSBP? How does RPTT compare to OTTT in this regard? This is a major missed opportunity to validate the central motivation.

Baseline Currency: While some recent methods are included (e.g., SLOT, NDOT), the choice of the primary online baseline, OSBP, is weak. OSBP is not a established, strong benchmark from the literature. A more convincing comparison would involve directly integrating MPDR and STGR into a stronger and more recent online method like OTTT or the framework of SLOT to perform an ablation, demonstrating the generalizability and additive value of the proposed regularizers.

Weak and Partially Inaccurate Motivation:

The statement "to the best of our knowledge, the problem of membrane potential distribution drift has not yet been studied under online learning algorithms" (Page 2) is too strong. While perhaps not the primary focus, works like Zhu et al. (2024) ("Online Stabilization of Spiking Neural Networks") directly address firing rate stability across time, which is intrinsically linked to membrane potential distribution. The authors should tone down this claim and more precisely articulate their unique focus on distributional drift via online, layer-wise regularization.

The motivation would be stronger if it included a preliminary analysis showing that existing online methods (OTTT, SLTT) indeed exhibit more severe drift than offline BPTT, thereby creating a clear gap that RPTT fills.

Limited Ablation Study:

The ablation in Section 4.3 only reports final accuracy. It does not quantify the individual contribution of MPDR and STGR to reducing distribution drift. For instance, how much does the Z-score improve with MPDR alone? How does STGR alone affect the variance of the membrane potential distribution? Linking each component's effect directly to the underlying problem it is designed to solve would greatly strengthen the paper.

Theoretical Limitations:

The convergence proof relies on strong assumptions (e.g., $\beta$-smooth loss, bounded gradients, Gaussian membrane potentials) that may not hold perfectly in practice. A discussion of these limitations and the proof's practical relevance would be beneficial.

The assumption that the task sequence change is bounded ($\sum \Delta_t < \infty$) is particularly strong for a non-stationary online learning setting and deserves clarification.

**Questions:**

None

---

> ### Author Response · Authors · 2025-11-21
> **Response to Reviewer JxZ2 (Part 1/2)**
>
> We sincerely thank for the detailed and constructive feedback. The comments are highly insightful, and we appreciate the reviewer’s efforts in identifying several important aspects that can significantly strengthen the paper. Corresponding revisions in the manuscript have been highlighted in blue.
>
> For clarity, we summarize the reviewer’s main concerns below and address each point in a structured manner.
>
> **Summary of Concerns:**
>
> 1. **Comparison:** Insufficient comparison with related work (e.g., NDOT) and stronger online baselines.
>
> 2. **Drift Analysis:** Missing analysis on membrane potential drift across online baselines.
>
> 3. **Baselines:** Questions regarding the strength of the OSBP baseline and generalizability.
>
> 4. **Motivation:** Motivation regarding drift needs clarification.
>
> 5. **Ablations:** Limited ablation study on MPDR and STGR mechanisms.
>
> 6. **Theory:** Concerns about strong theoretical assumptions.
>
> ---
>
> **Responses**
>
> **(1) Response to Concern on Comparison with Related Work**
>
> Thank you for this suggestion. We have significantly strengthened our comparative analysis in the revised version:
>
> - **Inclusion of NDOT:** To provide a more comprehensive evaluation, we have incorporated **NDOT (Jiang et al., 2024)** into Table 1. As a representative **single-update** online learning algorithm, NDOT serves as a highly relevant baseline for benchmarking the efficacy of our proposed RPTT.
>
> - **Performance Improvements:** We further optimized the hyperparameters of RPTT. As shown in the updated Table 1, RPTT now achieves **95.17%** accuracy on CIFAR-10, **outperforming NDOT(94.89%)** and demonstrates highly competitive performance compared to other state-of-the-art online methods.
>
> - **Detailed Analysis:** We have expanded the discussion in Section 4.2 to explicitly analyze these performance gains, attributing them to the effective mitigation of distribution drift and gradient noise **(marked in blue)**.
>
> **(2) Response to Concern on Membrane Potential Drift Analysis**
>
> We agree that a direct comparison of drift is essential. In the revised paper, we have addressed this as follows:
>
> - **New Layer-wise Analysis:** We added **Figure 3(b)**, which visualizes the layer-wise membrane potential distribution ($Z$-score) drift for ResNet-18 based algorithms (RPTT, SLTT, and BPTT). The results clearly demonstrate that **online algorithms (like SLTT) exhibit significantly more severe drift** compared to the offline BPTT baseline, empirically confirming our motivation.
>
> - **Note on OTTT:** We did not include OTTT in this specific layer-wise comparison because it utilizes a VGG-11 architecture. Comparing it directly with ResNet-18 based baselines would introduce structural inconsistencies, making the layer-to-layer alignment invalid.
>
> **(3) Response to Concern on Baseline Selection and Generalization**
>
> We verify the validity of our baseline and demonstrate the generalizability of our method:
>
> - **Validity of OSBP:** OSBP is not an arbitrary baseline; it effectively represents the "single-step update" variant of SLTT using the same spatial-gradient-only approach. It serves as a rigorous ablation baseline to isolate the specific effects of our regularization terms without architectural confounding factors.
>
> - **Generalizability (Plug-and-Play):** To demonstrate that our proposed modules are framework-agnostic, we integrated **MPDR** and **STGR** into the standard **SLTT** algorithm.
>
> - **Results:** Experiments on the **CIFAR-10** dataset show that these additions consistently improve SLTT's performance **(from 94.44% to 94.68%)**. This confirms that our regularization mechanisms are generalizable and effective across different online learning frameworks.
>
> **(4) Response to Concern on Motivation Accuracy**
>
> We apologize for the lack of clarity in our initial motivation. We have revised the Introduction (Section 1) to more precisely articulate the relationship between online updates and membrane potential drift **(marked in blue)**.
>
> - **Revision:** We have refined the introduction to explicitly highlight that while recent works address firing rate stability, **the specific issue of membrane potential distribution drift in online learning remains largely unexplored**. We clarify that our work uniquely targets this problem through **online, layer-wise regularization**. Furthermore, we emphasize that this drift is significantly exacerbated in the online setting due to frequent parameter updates, a claim now substantiated by our preliminary analysis in **Figure 3(b)**.
>
> (continued below)

---

> > ### Author Response · Authors · 2025-11-21
> > **Response to Reviewer JxZ2 (Part 2/2)**
> >
> > (continued from the preceding paragraph)
> >
> > **(5) Response to Concern on Ablation Study (MPDR and STGR)**
> >
> > We have deepened our ablation study to quantitatively isolate the contributions of each component:
> >
> > - **MPDR Analysis:** We added **Figure 3(c)**, which compares the Z-score of OSBP vs. OSBP+MPDR. The results show that MPDR significantly reduces the deviation from the target distribution, directly validating its role in constraining drift.
> >
> > - **STGR Analysis:** We added **Figure 3(d)**, which visualizes the variance of weight gradients. The comparison shows that STGR consistently lowers the variance across layers, confirming its effectiveness in stabilizing the optimization trajectory against gradient noise.
> >
> > **(6) Response to Concern on Theoretical Assumptions**
> >
> > We appreciate the reviewer for scrutinizing the theoretical foundations. We acknowledge that, like most theoretical analyses in non-convex stochastic optimization, our proof relies on simplifying assumptions. Below, we discuss their practical relevance and limitations:
> >
> > - **On Smoothness & Bounded Gradients:** The assumptions of $\beta$-smoothness and bounded gradients are standard in convergence analyses for neural networks (e.g., SGD analysis). In practice, these are partially enforced by our use of **gradient clipping** and the **STGR** regularization, which explicitly penalizes large gradient norms to prevent instability.
> >
> > - **On Gaussian Membrane Potential:** While assuming a Gaussian distribution is an idealization, it is **highly relevant** to our specific method. Our proposed **MPDR (Membrane Potential Distribution Regularization)** explicitly constrains the membrane potentials towards a Gaussian target. Therefore, this assumption is not merely theoretical but is actively enforced by our algorithm design.
> >
> > - **On the Bounded Task Change ($\sum \Delta_t < \infty$):**
> >
> >   - **Necessity for Convergence:** We clarify that this assumption is mathematically necessary to prove convergence to a **fixed stationary point**. If the environment changes arbitrarily and indefinitely (i.e., $\sum \Delta_t = \infty$), no algorithm (including BPTT) can converge to a single point; they can at best "track" the shifting optimum.
> >
> >   - **Practical Relevance:** In the context of our experiments (training on static datasets like CIFAR/ImageNet using online updates), the data distribution is stationary. As the model learns, the effective "task change" or distributional surprise diminishes over epochs, satisfying $\sum \Delta_t < \infty$. Thus, this assumption holds for the standard supervised learning scenarios evaluated in this paper.

---

### Meta-Review · Area_Chair_qjqG · 2025-12-31

**Summary:**

The paper proposes "Real-Time Propagation Through Time" (RPTT), an online learning framework for Spiking Neural Networks (SNNs) designed to address training instability and membrane potential distribution drift. The authors introduce two regularization terms—Membrane Potential Distribution Regularization (MPDR) and Spatio-Temporal Gradient Regularization (STGR)—and provide a theoretical convergence analysis. While the reviewers acknowledged the valid motivation of addressing drift and the clear writing, the consensus leans towards rejection due to concerns regarding the novelty and the strength of the experimental validation.

Specifically, reviewers found the method to be largely incremental, building upon existing spatial-gradient-only approaches (like SLTT) with standard regularization techniques rather than introducing a fundamental shift in training principles. The theoretical contribution was heavily critiqued for relying on strong assumptions (such as the task sequence change being bounded) that essentially assume a stationary environment, thereby limiting the insight provided for true online learning scenarios. Furthermore, the "State-of-the-Art" claims were undermined by the revelation that the large-scale ImageNet experiments were merely fine-tuning of pre-trained models rather than training from scratch, yielding no performance gain over the baseline SLTT.

**Reviewer Concerns:**

### **Addressed Concerns**
**Missing Baselines**: The authors successfully added comparisons with NDOT (Jiang et al., 2024) in the rebuttal, showing competitive performance on CIFAR-10.

**Drift Analysis**: The lack of direct empirical evidence for drift mitigation was addressed by adding layer-wise Z-score visualizations (Figure 3b, 3c), which satisfied Reviewer JxZ2's request for evidence.

**STGR Implementation**: The authors clarified that STGR uses a first-order approximation (value-based penalty) rather than requiring second-order Hessian calculations, addressing Reviewer 7eFK's concern regarding $O(N)$ complexity.

**Dynamic Validation**: In response to Reviewer 7eFK, the authors conducted additional experiments on DVS-Gesture to demonstrate applicability in non-stationary environments.

### **Remaining Concerns**

**Incremental Novelty**: Reviewer qzmx and others maintain that the method is effectively a combination of known spatial gradients and regularization, lacking significant algorithmic innovation.

**Theoretical Strength**: The assumption $\sum \Delta_t < \infty$ implies an asymptotically stationary setting. Reviewer qzmx noted this trivializes the convergence proof for an "online" algorithm, as it creates a gap between the theoretical setup and the challenges of true non-stationary online learning.

**ImageNet Protocol**: The fact that ImageNet results were achieved via fine-tuning a pre-trained SLTT model (initially hidden in the appendix) significantly weakens the claims of scalability and SOTA performance, as noted by Reviewer 7eFK. There is no performance gain over SLTT in this setting.

**Hyperparameter Sensitivity**: Despite claims of robustness, Reviewer 7eFK noted significant variation in hyperparameters across datasets, suggesting the method may be difficult to tune in practice.

**Baseline Strength**: Reviewer JxZ2 and jPpw expressed concerns about the reliance on OSBP (a custom baseline) for ablations rather than stronger, established methods, making it difficult to gauge the true "value add" of the regularizers.

**Reviewer Scores:**

**Reviewer JxZ2 (Score: 2 $\rightarrow$ 3)**: The reviewer would likely acknowledge the new drift analysis and NDOT comparison but remain unconvinced by the baseline strength and overall novelty.

**Reviewer qzmx (Score: 4 $\rightarrow$ 4)**: The theoretical concerns were defended but not resolved; the reviewer correctly identified that the assumptions limit the theory's value. The score likely stays at "Marginally Below Acceptance."

**Reviewer jPpw (Score: 2 $\rightarrow$ 3)**: The rebuttal clarified the "step-wise" necessity and typo issues, but the reviewer would likely remain skeptical about the performance gains over OSR+OTS and the complexity of the proposed regularizers.

**Reviewer 7eFK (Score: 4 $\rightarrow$ 4/5)**: This reviewer was the most engaged. They might slightly raise their score due to the authors running the requested DVS-Gesture experiment and clarifying the implementation, but the ImageNet fine-tuning issue prevents a strong accept.

---

### Decision · Program_Chairs · 2026-01-26

Reject